# Do private providers give patients what they demand, even if it is inappropriate? A randomised study using unannounced standardised patients in Kenya

Ada Kwan  ,[1,2] Claire E Boone,[2] Giorgia Sulis  ,[3] Paul J Gertler[4]

¹Department of Medicine, University of California San Francisco, San Francisco, California, USA
²Division of Health Policy and Management, University of California Berkeley School of Public Health, Berkeley, California, USA
³Department of Epidemiology, Biostatistics and Occupational Health, McGill University, Montreal, Canada
⁴University of California Berkeley Haas School of Business, Berkeley, California, USA

**Correspondence to**
Dr Ada Kwan;
ada.kwan@ucsf.edu

## ABSTRACT

**Introduction** Low and varied quality of care has been demonstrated for childhood illnesses in low-income and middle-income countries. Some quality improvement strategies focus on increasing patient engagement; however, evidence suggests that patients demanding medicines can favour the selection of resistant microbial strains in the individual and the community if drugs are inappropriately used. This study examines the effects on quality of care when patients demand different types of inappropriate medicines.

**Methods** We conducted an experiment where unannounced standardised patients (SPs), locally recruited individuals trained to simulate a standardised case, present at private clinics. Between 8 March and 28 May 2019, 10 SPs portraying caretakers of a watery diarrhoea childhood case scenario (in absentia) conducted N=200 visits at 200 private, primary care clinics in Kenya. Half of the clinics were randomly assigned to receive an SP demanding amoxicillin (an antibiotic); the other half, an SP demanding albendazole (an antiparasitic drug often used for deworming), with other presenting characteristics the same. We used logistic and linear regression models to assess the effects of demanding these inappropriate medicines on correct and unnecessary case management outcomes.

**Results** Compared with 3% among those who did not demand albendazole, the dispensing rate increased significantly to 34% for those who did (adjusted OR 0.06, 95% CI 0.02 to 0.22, p<0.0001). Providers did not give different levels of amoxicillin between those demanding it and those not demanding it (adjusted OR 1.73, 95% CI 0.51 to 5.82). Neither significantly changed any correct management outcomes, such as treatment or referral elsewhere.

**Conclusion** Private providers appear to account for both business-driven benefits and individual health impacts when making prescribing decisions. Additional research is needed on provider knowledge and perceptions of profit and individual and community health trade-offs when making prescription decisions after patients demand different types of inappropriate medicines.

**Trial registration numbers** American Economic Association Registry (#AEARCTR-0000217) and Pan African Clinical Trial Registry (#PACTR201502000770329).

### Strengths and limitations of this study

► Use of the standardised patient (SP) method, where locally recruited individuals are trained to present the same scenario, offers the ability to compare provider practice across a sample of private health providers who are faced with a patient who demanded inappropriate care.
► This study uses a randomised design to causally determine relative differences in provider behaviour when two inappropriate medicines are demanded by unannounced SPs with random assignment, with the presenting scenario the same otherwise.
► A limitation of the SP method is that the findings do not fully reflect real patient behaviour.
► This study is not able to assess the level of provider awareness regarding the appropriateness of each medicine or the condition under investigation.

## INTRODUCTION

Individuals seeking healthcare services sometimes demand inappropriate medicines, such as antibiotics, based on the widespread misperception that this would lead to faster and better recovery.[1 2] Regardless of adequate training and knowledge of clinical practice guidelines, providers may grant these requests to facilitate patient satisfaction and to avoid negative judgments.[3] For-profit providers may be concerned that these negative judgments, and their overall reputation, can reduce the likelihood of patients returning for subsequent visits, which can affect their bottom line. Prescribing behaviours that arise from these concerns may vary based on the extent to which medicines demanded are harmful or perceived as such and, in the private sector, profitable.[4–8] In this paper, we study the effects of patients demanding two different inappropriate medicines, as examples of trade-offs

providers might make between risks, profits, and patient satisfaction.[6 7 9 10]

These dynamics pertain to policy. Understanding the relationship between inappropriate dispensing behaviours and what patients demand from providers is important for designing quality improvement interventions. Public health authorities and many studies cite the overuse and misuse of antimicrobials as the main drivers of drug-resistance.[11] However, there is at best limited literature on the effects of when patients demand medicines on provider prescribing behaviour in low/middle-income countries (LMICs).[6 7] On the patient side, studies in high-income countries suggest that patient and provider knowledge, attitudes and expectations are important drivers of antibiotic prescriptions.[12–14] One example is the notion of patient activation, or when 'patients who have the *motivation, knowledge, skills, and confidence* to make *effective* decisions to manage their health' (emphasis ours).[15] Patient activation has been extensively studied in the USA, and this research emphasises the potential for interventions that increase informed and 'active' patients, particularly because of its association with better health and healthcare outcomes.[16 17] However, patient activation is different from when patients demand antibiotics that are inappropriate for their conditions. Further, many of these studies report associations and cannot differentiate whether increased engagement results in increased quality of care or the reverse. Thus, constructing effective interventions on patient engagement becomes challenging if actors or mechanisms for intervening to improve care are unclear. This suggests that the patient's role could have a larger influence on better care relative to the provider. That patients can have a larger influence on services begs the question, *What happens when patients demand different types of inappropriate care?*

Our study's objective is to examine the role of patient demand for inappropriate care, on prescribing and dispensing practices for childhood diarrhoea in Kenya. The government maintains explicit guidelines for childhood diarrhoea case management (see online supplemental appendix A1),[18] and we use the standardised patient (SP) method, which provides an unbiased way to compare multiple providers because of a standardised case scenario presentation, to measure care levels. We draw on childhood diarrhoea for several reasons. First, several studies have validated the use of the SP method for examining childhood diarrhoea, including in Kenya.[19–23] Over other existing quality of care methods, the SP method has many advantages and also controls for patient mix and sorting. For example, provider surveys measure provider knowledge rather than actual practice; exit interviews suffer from recall bias and clients may also not be able to discern specific clinical actions; providers may perform differently under direct observation, known as the Hawthorne effect; and in these settings, the quality of administrative data or records is often varied, if it exists at all.[8 23] Accruing evidence from SP studies on childhood diarrhoea across LMICs demonstrate that quality of care

is low and varied for correct management of childhood illnesses.[19 20 24–29]

Second, although the global burden of diarrhoeal disease is declining over time, it remains a major concern in LMICs where poor sanitation and hygiene along with indiscernibly varied quality healthcare make this health condition among children common and often life-threatening. With 1.73 billion episodes a year, diarrhoea remains one of the leading causes of child morbidity and mortality worldwide.[30 31]

Third, diarrhoea is an interesting condition to examine the role of patient demand on appropriate and inappropriate care. Diarrhoea is defined as an increase in frequency of bowel movements (usually three or more per day), accompanied by a decrease in stool consistency.[32] Although a wide range of pathogens can cause diarrhoeal disease, consumption of contaminated food or water and interpersonal contacts in poor hygiene conditions constitute a common denominator. Rotavirus, *Escherichia coli*, *Cryptosporidium* spp and *Shigella* spp are the most common causal agents in lowest-income settings.[33] The WHO Integrated Management for Childhood Illness handbook was published in 2005 to provide a structured and simplified approach to the assessment and therapeutic management of children presenting with various clinical pictures in first-level primary care facilities, particularly in resource-limited areas.[34] With respect to diarrhoea, antimicrobial treatment is only recommended under selected circumstances (eg, evidence of blood in the stool).

This study contributes to the literature in several ways. First, this study adds to the understanding of how pervasive is overprescription. Recent studies on health conditions, including common childhood illnesses, in LMICs show that the overuse of medicines pose dangers of resistance that have individual and public health level consequences.[24 25 35] Second, we provide experimental evidence showing that patient-related determinants influence appropriate and inappropriate treatment.[1 2 36] Two other studies to our knowledge examine the effects of patient demand specifically on the rates of antibiotic dispensing with the SP method and find that (i) SPs who share knowledge that antibiotics are inappropriate in China were less likely to receive antibiotics and (ii) antibiotic prescription rates reduced when SPs demanded them alongside a statement that they would make the purchase elsewhere.[6 7] In an LMIC setting, which is under-represented in the literature on this topic, this study additionally extends the current literature on the role of caregivers demanding two types of antimicrobial medicines for a condition that, for most children with this condition, requires only supportive treatment. The majority of diarrhoea cases do not need microbial therapy and only require supportive treatment, such as rehydration.[32 34] Third, this study can help inform governments that are committed to universal access to high quality of care worldwide. Understanding how quality of care can be improved is critical, particularly in the private sector in countries where a substantial amount of care is provided by the private sector.[37] This

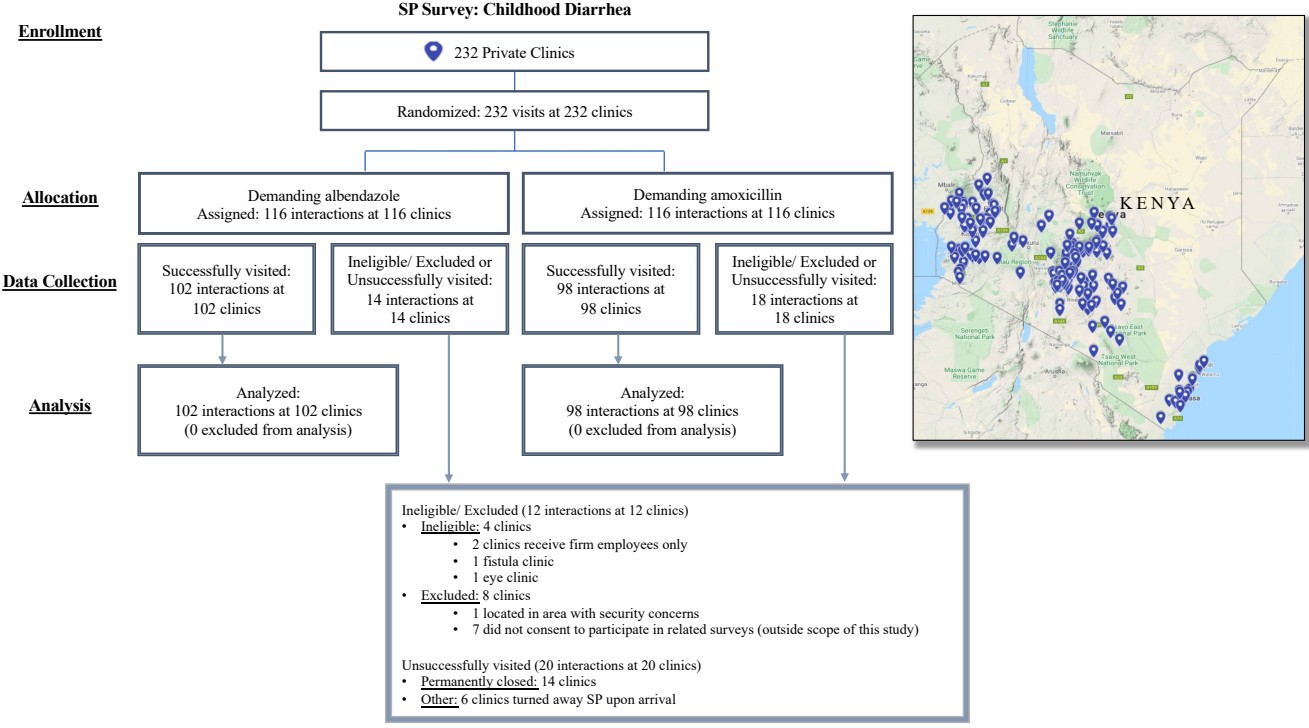

**Figure 1** Clinic sample and SP randomised study design. SP, standardised patient.

study contributes to the knowledge of quality improvement mechanisms by understanding the provision of support treatment, as well as the provision of unnecessary or potentially harmful treatment at point of care.

## METHODS
This study examines SP and provider vignette data collected in 2019 across 200 private clinics spread across 35 of Kenya's 47 counties. Figure 1 shows a map of Kenya and a Consolidated Standards of Reporting Trials diagram for clinic sampling. For this study, we exploited the private clinic sample (N=232) for the impact evaluation of a programme called African Health Markets for Equity (AHME) and excluded clinics that were ineligible to receive walk-ins for childhood illnesses,[4] were located in an area with security concerns,[1] did not consent to the AHME evaluation study,[7] or were permanently closed.[14] We did not capture data from six clinics, which turned away the SP on arrival. The programme is not the focus of this study, but additional details related to the programme and clinic sample are provided in online supplemental appendix A2 when relevant for this study, including AHME assignment across demanding arms (online supplemental table A1).

## Data sources
Between 8 March and 28 May 2019, 200 unannounced SP visits were completed at 200 private Kenyan clinics. Data were captured at two moments during the interaction: 'predemanding' includes actions before the SP demanded

the assigned medicine, and 'postdemanding' includes all actions by the completion of the visit. We analyse N=200 predemanding and N=200 postdemanding observations for a childhood diarrhoea case scenario. Using Stata V.16 (StataCorp), half the clinics were randomly assigned to receive an SP demanding albendazole, and the other half to receive an SP demanding amoxicillin. SP requests were done at the end of the visit or earlier only if it was necessary to avoid an unusual visit.

The scenario represents a 28-year-old mother who comes to the clinic with a 1.5-year-old child at home sick with watery diarrhoea (see table 1). If probed by the provider, the SP is trained to share that the child is a little hot and has passed approximately 6–7 stools in the last 2 days. This study follows the design and protocol with sample size calculations based on the childhood diarrhoea SP case scenario described in Daniels *et al*.[22] In our study, the SP visits were conducted by 10 females locally recruited, trained and hired as SPs. All SPs were seemingly healthy, so providers would not detect and treat other health ailments that were unrelated to the presenting scenario. All data reflect quality measures for SPs seeking walk-in, outpatient services. Online supplemental appendix A3 contains additional details on SP case development, recruitment, training, piloting and sample size calculations (online supplemental table A2).

The SP method minimises bias in assessing provider practice. To assess care provided to patients in LMIC settings, the literature describes several methodologies, including direct observation, administrative or medical

**Table 1** Description of childhood diarrhoea standardised patient (SP) case scenario and main outcomes

| Case | Case description | SP experiments varying patient characteristics | Main outcomes |
|------|------------------|-----------------------------------------------|---------------|
| Childhood diarrhoea | A 28-year-old mother comes to the clinic with a 1.5-year-old child at home sick with acute watery diarrhoea. The child is a little hot and has passed approximately 6–7 stools in the last 2 days | *Experiment 1:* demanding albendazole | **Correct case management (=1):** any one of the following were done by the provider: gave ORS, advised on ORS, referred elsewhere, asked to return to clinic for any reason |
| | | *Experiment 2:* demanding amoxicillin | **Any unnecessary medicines (=1):** any medicines given excluding ORS, zinc |

ORS, oral rehydration salts.

record abstraction, client exit interviews, provider vignettes and SPs. Each method has its own interpretation and set of advantages and disadvantages which is described at length elsewhere.[8 38 39] To identify the effect of what happens when a patient demands an inappropriate antibiotic or antiparasitic medicine, we randomly assigned whether the SP would demand amoxicillin or albendazole, respectively. Importantly, both medications are considered unnecessary for a child with watery diarrhoea. Both medications also have harmful effects for the community if systematically and unnecessarily used. Online supplemental appendix A3.3 includes details on how these two medicines were selected. The SP method has the advantage that the researchers know the true condition of the 'patient' which is not possible when examining data derived from real patients. SP data particularly allow for providers across different facilities to be compared against the exact same patient scenario and is thus increasingly considered the gold standard for measuring provider practice across a sample of providers that lack standardised health records.

To ensure accurate and comprehensive recall, within 1–3 hours after each SP visit, SPs completed an exit questionnaire administered by a fieldwork supervisor. The exit questionnaire collected information regarding the SP's visit, including time of arrival, time of departure, history questions asked, diagnosis, lab tests ordered, medicines dispensed and prescribed, counselling given, and a subjective assessment of the visit. Further, for each visit, SPs and their supervisors attempted to identify all providers seen by the SPs. The list of providers formed the provider survey sampling frame.

Literature on quality of care in LMICs shows that large differences exist between what healthcare providers know and do.[40] To measure whether providers know what to do in this setting, we additionally analyse data from the provider survey conducted between November and December 2019 among providers who saw SPs. The provider survey included a vignette module to assess knowledge based on a childhood diarrhoea vignette case matching the SP case scenario, in addition to capturing provider characteristics through interview.

**Outcomes**

Using SP and vignette data, we constructed binary measures for our main outcomes of interest: correct case management and whether any unnecessary medicines were prescribed or dispensed, since one aspect of quality of care is not only dispensing correct medicines but also *not* dispensing inappropriate medicines. Benchmarked against national guidelines, correct case management refers to the minimal and essential actions for childhood watery diarrhoea case management (see online supplemental figure A1).[18] Visits were coded as being correctly managed (=1) if the provider did any one of the following: gave oral rehydration salts (ORS), advised on ORS, referred the SP or asked the SP to return; 0, otherwise (table 1). We classified ORS and zinc to be appropriate, and a provider was coded as ordering any unnecessary medicines if others were prescribed or dispensed. We define 'prescribe/dispense' as a term to capture the provider's intention to give a medicine to the patient, regardless of whether the SP walked away with it: 'prescribe' captured a situation where the provider may have written a prescription, including when the SP may not have actually received it (eg, a stockout); 'dispense' captured a situation where the provider may have given the medicine, including when the provider may not have written a prescription. We examine whether the provider prescribes/dispenses amoxicillin or albendazole. Additionally, whether any antibiotic and/or any antiparasitic (including antimalarials) were assessed.

**Statistical analysis**

We first conducted difference-in-means tests on clinic characteristics uncorrelated as a balance check to confirm the random assignment of demanding experiments to clinics were balanced. Next, we computed adjusted ORs (aORs) with 95% CIs from a logistic regression model, while controlling for differences that arose from our design, including a binary SP experiment variable (0 if the SP was assigned to demand albendazole; 1 if assigned to demand amoxicillin); the binary AHME treatment indicator representing whether the clinic associated with the SP visit received the AHME intervention (=1) or was

assigned to the control arm (=0), which was randomised independently from the SP experiment (see online supplemental table A1 for AHME assignment); and fixed effects at the SP individual level, as illustrated in previous SP studies with similar designs. The parameter of interest is the SP experiment coefficient which is interpreted as the effect of demanding amoxicillin relative to albendazole on the outcome of interest. We complemented these analyses with ordinary least squares regression to assess differences in outcomes across the demanding experiments. It is important to note that our estimates correspond with the expected average quality of care and demanding differences if the clinics were selected randomly by a patient in the country.

Analyses using SP data were conducted at the SP-provider visit level. When SP data were linked to provider survey data, the unit of observation is a successful (ie, completed) SP-provider visit with provider survey responses from the provider seen during the SP visit. All data analyses were performed with Stata V.16, and deidentified interaction data with variables and code needed to re-create the results reported in this article are available.[41]

### Patient and public involvement
Health care providers from Kenya who were hired as technical advisors were involved in advising on the definition of outcomes. Individuals recruited locally and trained to be SPs were involved in the case design and data collection fieldwork for this research (additional details in online supplemental appendix A3).

### RESULTS
A total of 200 unannounced SP-provider visits were successfully conducted by 10 SPs at a total of 200 different private health clinics across 34 of the 47 counties in Kenya (figure 1, additional fieldwork details in online supplemental appendix A5). To ensure that the experiment was successfully randomised, we checked differences in means for the clinics assigned an SP demanding amoxicillin or albendazole across various characteristics (table 2). Since the groups randomly assigned to receive SPs demanding different medicines are balanced on data from the year SP visits were conducted (ie, the absolute difference between the mean value in the two groups is not different from zero), we can rely on our statistical model assumption that the randomisation of demanding assignments created exchangeable treatment arms to assess the impacts of demanding different unnecessary drugs. Thus, we can interpret our coefficient of interest as an unbiased estimate of the effect of demanding each medicine on our outcomes.

Table 3 provides summary statistics for the N=200 SP postdemanding observations. Nearly 40% of providers were female and just over half of the visits were conducted with a provider that appeared between 30 and 50 years of age. More SPs saw a medical doctor or clinical officer (33%) than other provider types, followed by a nurse or midwife (30%). On average, there were approximately 1.55 individuals waiting in the waiting room when the SP arrived (to capture how busy the clinic was, in lieu of utilisation data), and each SP visit lasted 6.65 min with the provider, who asked on average 4.46 history questions. Among the visits, 15% resulted in a correct diagnosis or suspicion of watery diarrhoea, and 75% of the visits were correctly managed with 31% of SPs asked to return and a very small percentage (6%) referred elsewhere. Despite 75% of the SP visits being correctly managed in practice, 90% of the visits had a provider who knew how to correctly manage the case as measured in the administered provider vignette (see online supplemental figure B1 for more comparisons across knowledge and practice). Because outcomes that were captured before demanding ('predemanding') cannot be entirely interpreted as a complete interaction, we only report postdemanding measures (see online supplemental figures B2 and B3 for predemanding outcomes).

### Effects of demanding on levels of correct and unnecessary services
Figure 2 reports aORs comparing demanding albendazole versus demanding amoxicillin across various binary quality of care outcomes, adjusting for the AHME treatment assignment and SP individual fixed effects. We did not find that the type of unnecessary medicine demanded had an estimated effect on correct case management or any of its components (advising on ORS, giving or advising on ORS, asking to return, or referring the patient for any reason). However, the aOR of being dispensed or prescribed zinc, which is advised within the minimum package for facility case management as per the Kenya national guidelines because of its benefits for reducing duration and severity of episodes for watery diarrhoea,[18] was 1.92 (95% CI 0.96 to 3.86; p=0.066) for those who demanded amoxicillin, relative to those who demanded albendazole. Though not statistically significant at the 5% level, this difference has a clinical significance since the lower bound of the 95% CI is very close to 1. Despite how zinc is often recommended in addition to ORS to shorten the duration of symptoms, it is not mentioned in the guidelines to be available at private health facilities. Regardless, those who demanded albendazole were 33.0% less likely to receive zinc supplementation (coefficient=−0.148, SE=0.081, p=0.071; online supplemental table B2A, column 8).

With respect to inappropriate medicines, demanding albendazole significantly favours the odds that albendazole is dispensed/prescribed, relative to the visits where the SP demanded amoxicillin (aOR in favour of SPs demanding amoxicillin: 0.06, 95% CI 0.02 to 0.22, p<0.0001). This translates into a 34.8 percentage point significant increase (SE 0.059, p<0.001; online supplemental table B2A, column 13) in whether albendazole was given, compared with 3.1% of SPs who did not demand albendazole receiving it. This effect is similar for whether any antiparasitic is dispensed/prescribed (aOR in favour

**Table 2** Balance across characteristics of clinics assigned albendazole versus amoxicillin demanding experiment

| | Clinics assigned to receive an SP demanding albendazole, n=102 | | | Clinics assigned to receive an SP demanding amoxicillin, n=98 | | | |
|---|---|---|---|---|---|---|---|
| | N | Mean | 95% CI | N | Mean | 95% CI | P value |
| Hours open per week | 94 | 101.86 | (91.77 to 111.96) | 88 | 94.91 | (85.01 to 104.81) | 0.337 |
| Average of hours open per day | 94 | 14.85 | (13.46 to 16.24) | 88 | 14.12 | (12.79 to 15.45) | 0.460 |
| Clinic is NHIF empaneled | 94 | 0.30 | (0.21 to 0.39) | 88 | 0.32 | (0.22 to 0.42) | 0.768 |
| Number of clients | 94 | 466.86 | (353.84 to 579.89) | 88 | 525.27 | (354.13 to 696.42) | 0.573 |
| *Data missing* | 102 | 0.08 | (0.03 to 0.13) | 98 | 0.10 | (0.04 to 0.16) | 0.562 |
| Count of total staff | 95 | 3.97 | (3.30 to 4.64) | 88 | 3.99 | (3.38 to 4.60) | 0.965 |
| Count of clinical staff (doctors and nurses) | 95 | 2.23 | (1.81 to 2.65) | 88 | 2.20 | (1.86 to 2.55) | 0.923 |
| *Data missing* | 102 | 0.07 | (0.02 to 0.12) | 98 | 0.10 | (0.04 to 0.16) | 0.400 |
| Facility has community health workers | 94 | 0.40 | (0.31 to 0.50) | 88 | 0.33 | (0.23 to 0.43) | 0.299 |
| Total revenues (USD) | 92 | 4534.70 | (2493.57 to 6575.83) | 85 | 3621.63 | (2106.29 to 5136.97) | 0.488 |
| Total profits (USD) | 91 | 1487.57 | (570.25 to 2404.89) | 83 | −2665.31 | (−8670.93 to 3340.31) | 0.163 |
| Total expenditures (USD) | 92 | 3057.43 | (1548.01 to 4566.85) | 85 | 6175.33 | (−831.09 to 13 181.75) | 0.378 |
| Services provided at clinic | | | | | | | |
| *Facility provides any curative services* | 94 | 0.97 | (0.93 to 1.00) | 88 | 0.95 | (0.91 to 1.00) | 0.637 |
| *Antenatal care* | 94 | 0.69 | (0.60 to 0.79) | 88 | 0.68 | (0.58 to 0.78) | 0.889 |
| *Cervical cancer screening* | 94 | 0.49 | (0.39 to 0.59) | 88 | 0.48 | (0.37 to 0.58) | 0.871 |
| *Delivery* | 94 | 0.40 | (0.31 to 0.50) | 88 | 0.48 | (0.37 to 0.58) | 0.324 |
| *Dental services* | 94 | 0.16 | (0.09 to 0.23) | 88 | 0.17 | (0.09 to 0.25) | 0.844 |
| *Family planning* | 94 | 0.98 | (0.95 to 1.01) | 88 | 0.99 | (0.97 to 1.01) | 0.602 |
| *Imaging services (X-ray, ultrasound)* | 94 | 0.14 | (0.07 to 0.21) | 88 | 0.15 | (0.07 to 0.22) | 0.857 |
| *Immunisations visit* | 94 | 0.37 | (0.27 to 0.47) | 88 | 0.45 | (0.35 to 0.56) | 0.263 |
| *Inpatient services* | 94 | 0.26 | (0.17 to 0.34) | 88 | 0.26 | (0.17 to 0.35) | 0.926 |
| *Laboratory services* | 94 | 0.93 | (0.87 to 0.98) | 88 | 0.92 | (0.86 to 0.98) | 0.898 |
| *Malaria testing/treatment* | 94 | 0.96 | (0.92 to 1.00) | 88 | 0.94 | (0.90 to 0.99) | 0.659 |
| *Optical services* | 94 | 0.09 | (0.03 to 0.14) | 88 | 0.10 | (0.04 to 0.17) | 0.693 |
| *Pharmacy services* | 94 | 0.32 | (0.22 to 0.41) | 88 | 0.42 | (0.32 to 0.52) | 0.158 |
| *Postnatal care* | 94 | 0.55 | (0.45 to 0.65) | 88 | 0.58 | (0.48 to 0.68) | 0.722 |
| *Respiratory tract infections* | 94 | 0.98 | (0.95 to 1.01) | 88 | 1.00 | (1.00 to 1.00) | 0.171 |
| *Well-baby visit* | 94 | 0.62 | (0.52 to 0.72) | 88 | 0.62 | (0.52 to 0.73) | 0.912 |
| *Services—data missing* | 102 | 0.08 | (0.03 to 0.13) | 98 | 0.10 | (0.04 to 0.16) | 0.562 |

Number of observations refers to the number of clinics in the sample visited by SPs. The data source for this table does not have data available for all 200 private clinics in the sample. Data missing varies by type of variable—see 'data missing' for percentage of clinics where data are missing for number of clients, count of staff and services provided at the clinic.
AHME, African for Health Markets for Equity; NHIF, National Hospital Insurance Fund; USD, US dollar.

of SPs demanding amoxicillin: 0.18, 95% CI 0.07 to 0.43, p=0.0001).

We find higher prescribing/dispensing rates of any antibiotic relative to any antiparasitic (56% vs 25%, respectively; table 3) with 21% of visits resulting in both types of drugs being given. For all visits regardless of what the SP demanded, the most frequently given antibiotics were metronidazole (N=54, 27%), sulfamethoxazole and trimethoprim (N=38, 19%), metronidazole benzoate (N=24, 12%) and amoxicillin (N=19, 10%). We

find evidence that demanding amoxicillin has no effect on whether providers dispense/prescribe it (aOR: 1.73, 95% CI 0.51 to 5.82, p=0.3778) with a similar null finding on whether providers dispense/prescribe any antibiotic (aOR: 0.94, 95% CI 0.48 to 1.84, p=0.8526) relative to the visits with SPs demanding albendazole. Demanding albendazole versus amoxicillin resulted in different types of medicines being dispensed/prescribed at different frequencies across the SP visits (see online supplemental table B3).

**Table 3** Summary statistics of SP visits

| | (1) Pooled SP visits, n=200 | | (2) SP visits demanding albendazole, n=102 | | (3) SP visits demanding amoxicillin, n=98 | | (3)–(2) difference in means t-test |
|---|---|---|---|---|---|---|---|
| | N | Mean | N | Mean | N | Mean | P value |
| **Provider characteristics** | | | | | | | |
| Provider is female | 196 | 0.38 | 99 | 0.35 | 97 | 0.41 | 0.399 |
| Provider age group | 200 | | 102 | | 98 | | |
| *Under 30* | 33 | 0.17 | 18 | 0.18 | 15 | 0.15 | |
| *Between 30 and 50* | 114 | 0.57 | 59 | 0.58 | 55 | 0.56 | |
| *Above 50* | 42 | 0.21 | 18 | 0.18 | 24 | 0.24 | |
| *Missing data* | 11 | 0.06 | 7 | 0.07 | 4 | 0.04 | |
| Provider qualification | 200 | | 102 | | 98 | | |
| *Medical doctor or clinical officer* | 66 | 0.33 | 36 | 0.35 | 30 | 0.31 | |
| *Nurse or midwife* | 60 | 0.30 | 31 | 0.30 | 29 | 0.30 | |
| *Other staff* | 16 | 0.08 | 8 | 0.08 | 8 | 0.08 | |
| *Unknown or missing data* | 58 | 0.29 | 27 | 0.26 | 31 | 0.32 | |
| **Knowledge of correct management** | | | | | | | |
| *Diarrhoea* | 140 | 0.90 | 72 | 0.92 | 68 | 0.88 | 0.502 |
| **Visit characteristics** | | | | | | | |
| *Number of patients waiting when SP arrived* | 200 | 1.55 | 102 | 1.25 | 98 | 1.87 | 0.122 |
| *Minutes spent with provider* | 200 | 6.65 | 102 | 6.21 | 98 | 7.10 | 0.089 |
| *Number of history questions asked (post)* | 200 | 4.46 | 102 | 4.41 | 98 | 4.50 | 0.820 |
| *Correct diagnosis or suspicion (post)* | 200 | 0.15 | 102 | 0.11 | 98 | 0.19 | 0.089 |
| *Correct case management (post)* | 200 | 0.75 | 102 | 0.75 | 98 | 0.76 | 0.871 |
| *Any lab tests ordered (post)* | 200 | 0.13 | 102 | 0.16 | 98 | 0.10 | 0.251 |
| *Total lab tests ordered (post)* | 200 | 0.26 | 102 | 0.29 | 98 | 0.21 | 0.408 |
| *Any unnecessary lab tests (post)* | 200 | 0.10 | 102 | 0.08 | 98 | 0.12 | 0.302 |
| *Total unnecessary lab tests (post)* | 200 | 0.14 | 102 | 0.16 | 98 | 0.12 | 0.519 |
| *Number of medicines* | 200 | 2.38 | 102 | 2.40 | 98 | 2.35 | 0.845 |
| *Number of non-efficacious medicines* | 200 | 1.63 | 102 | 1.75 | 98 | 1.50 | 0.260 |
| *Dispensed/prescribed: albendazole* | 200 | 0.19 | 102 | 0.34 | 98 | 0.03 | 0.000 |
| *Dispensed/prescribed: antiparasitics* | 200 | 0.25 | 102 | 0.35 | 98 | 0.13 | 0.000 |
| *Dispensed/prescribed: amoxicillin* | 200 | 0.10 | 102 | 0.08 | 98 | 0.11 | 0.417 |
| *Dispensed/prescribed: antibiotics* | 200 | 0.56 | 102 | 0.56 | 98 | 0.55 | 0.912 |
| *Dispensed/prescribed: antibiotics and antiparasitics* | 200 | 0.21 | 102 | 0.28 | 98 | 0.12 | 0.004 |
| *Asked to return (post)* | 200 | 0.31 | 102 | 0.32 | 98 | 0.29 | 0.564 |
| *Referred elsewhere* | 200 | 0.06 | 102 | 0.07 | 98 | 0.05 | 0.602 |
| *Providers did good job explaining* | 189 | 0.76 | 95 | 0.74 | 94 | 0.79 | 0.419 |

Table displays summary statistics (N, mean) for all SP visits pooled (column 1), all SP visits assigned to demand albendazole (column 2) and all SP visits assigned to demand amoxicillin (column 3). Statistics with 'post' are postdemanding measures; all others are one time at the end of the visit. All summary statistics except knowledge of correct management for diarrhoea come from SP surveys. Knowledge of correct management is defined in the same way as correct case management and come from a vignette administered in the provider survey. Vignette data are matched to SP data for each SP visit by provider seen by SP or a replacement for the sampled provider. Provider age group is the estimated age group as perceived by the SP.

SP, standardised patient.

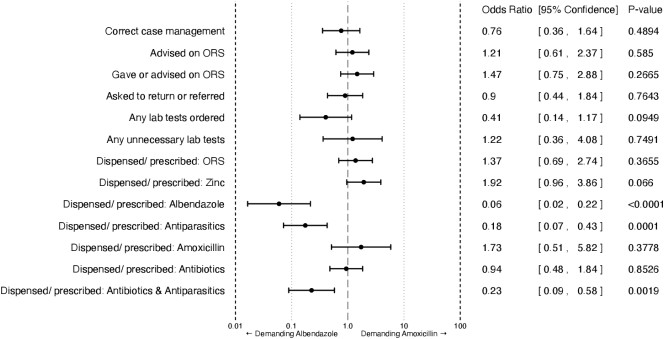

| | Odds Ratio | [95% Confidence] | P-value |
|---|---|---|---|
| Correct case management | 0.76 | [ 0.36 , 1.64 ] | 0.4894 |
| Advised on ORS | 1.21 | [ 0.61 , 2.37 ] | 0.585 |
| Gave or advised on ORS | 1.47 | [ 0.75 , 2.88 ] | 0.2665 |
| Asked to return or referred | 0.9 | [ 0.44 , 1.84 ] | 0.7643 |
| Any lab tests ordered | 0.41 | [ 0.14 , 1.17 ] | 0.0949 |
| Any unnecessary lab tests | 1.22 | [ 0.36 , 4.08 ] | 0.7491 |
| Dispensed/ prescribed: ORS | 1.37 | [ 0.69 , 2.74 ] | 0.3655 |
| Dispensed/ prescribed: Zinc | 1.92 | [ 0.96 , 3.86 ] | 0.066 |
| Dispensed/ prescribed: Albendazole | 0.06 | [ 0.02 , 0.22 ] | <0.0001 |
| Dispensed/ prescribed: Antiparasitics | 0.18 | [ 0.07 , 0.43 ] | 0.0001 |
| Dispensed/ prescribed: Amoxicillin | 1.73 | [ 0.51 , 5.82 ] | 0.3778 |
| Dispensed/ prescribed: Antibiotics | 0.94 | [ 0.48 , 1.84 ] | 0.8526 |
| Dispensed/ prescribed: Antibiotics & Antiparasitics | 0.23 | [ 0.09 , 0.58 ] | 0.0019 |

**Figure 2** Differences in quality of care by standardised patients (SPs) demanding albendazole versus amoxicillin. The chart illustrates estimated differences by the SP demanding experiment across quality-of-care outcomes. ORs are estimated controlling for SP fixed effects. All variables are binary outcomes. ORS, oral rehydration salts.

## DISCUSSION
### Main findings
Using the SP method, this study reports the extent to which provider treatment behaviours are influenced by patient demand for treatment, particularly two medicines that are unnecessary, harmful to the community, yet provided as empiric treatment for acute childhood watery diarrhoea. We compared the impact of a patient demanding an antibiotic medicine (amoxicillin, which has known public health risks for the individual and community), to an antiparasitic medication (albendazole) which is perceived to be harmless to the individual but also poses a risk to the community.

Our findings do suggest that providers who receive a client demanding amoxicillin are not likely to dispense what is demanded, which is not the case when a patient demands albendazole. Nonetheless, it is worth noting that, irrespective of patient demanding, 56% of the total 200 SP-provider visits carried out in our study were given or prescribed antibiotics, which is consistent with high rates from private facilities in other settings, such as India and Tanzania.[24 25 35] However, this proportion is higher than in the public sector in Kenya, as observed in a smaller cross-sectional SP study carried out in purposively sampled health facilities in Nairobi, where 32.5% (95% CI 20.0 to 47.5) of 40 SP-provider visits for child diarrhoea led to antibiotic prescribing.[22 24] Similar to other observations from African countries including Kenya, top prescribed antibiotics were from the WHO Access group, such as amoxicillin and metronidazole, partly reflecting the lower cost and easier access compared with other antibiotics.

### Strengths and weaknesses of the study
Our study has several strengths and weaknesses. First, the SP method is a particular method that requires a one-time visit for services that: do not subject the client to invasive procedures, can only assess tracer health conditions that have been validated for ethical research, and do not require established client services or follow-up visits, such

as those related to chronic conditions or other ailments. However, we identified these attributes of the method as favourable conditions to assess the quality of a walk-in outpatient service for child health services.

Second, we cannot compare levels of services without demanding, because by design, the SPs demanded medicines at the end of the visit, though on some occasions the SPs had to demand earlier (eg, when the provider was discussing treatment or sending the SP to the clinic's pharmacy) to ensure that there was a consistent and standardised narrative for the visit. A future study that seeks to compare demanding outcomes to not demanding is encouraged to implement separate SP visits. Here, it is important to recognise that SPs are not real patients and can behave in a way that confirms the study hypotheses which has been discussed in previous SP studies.[6 7 42] However, if this were the case for our study, the effects would likely be non-differential with respect to the type of medicine being demanded. Instead, an increased rate is only observed after demanding albendazole, and not after demanding amoxicillin.

Third, given that we only examine the interaction between providers and SPs, we do not report on the role of care-seeking behaviour and thus interpret findings conditional on patients seeking care. Further, since SPs are not real patients, what was found with SPs may not exactly reflect what happens with real patients, nor are we able to report on how satisfied real patients would have been given these prescription patterns. Similar to discussions in audit studies on discrimination, provider behaviours captured in this study as a response to SP features or trained characteristics may not translate to actual practice behaviours with real patients.[43 44] Our study does not conduct a detection survey to measure the extent of provider suspicion, but other SP studies with detection surveys find very low detection rates (0%–5%).[8 24 45] In the study where we based our childhood diarrhoea SP case, Daniels et al[22] administered a structured questionnaire 2 weeks after the completion of SP fieldwork in Nairobi, Kenya and found that despite providers having detected SPs in nine instances, none of these actually matched the study's SP visits. As described earlier, what the SP method allows us to do which other methods cannot is to identify what happens across multiple providers when providers are presented with SPs randomly assigned to demand different inappropriate medicines, with the same presentation otherwise.

Given these limitations, we have scrutinised a few possible channels in our data that are related to either provider behaviour or limitations of the method we implemented. Based on the study design, the increase in correct case management for watery diarrhoea can only be related to variables where we have captured the outcome at two time points: predemanding and postdemanding. Thus, effects of demanding on correct case management is related to advising on ORS or asking the patient to return. It cannot be from dispensing/prescribing ORS or referring to another facility, which are both captured

once at the end of the visit. One can imagine that having a predemanding and a postdemanding time point for each visit alerts us to an issue that the postdemanding environment simply captures more dispensing/prescribing of ORS, and thus higher correct care, because it captures all actions after the entire visit has been completed.

## Strengths and weaknesses in relation to other studies

This study adds to the literature in several ways. First, we extend the research done on the roles of patients and providers in the patient–provider relationship, in particular, what happens when patients demand inappropriate care and to what extent do demanding patients have an influence on services? Most notably, our findings stand in contrast with Currie et al's[7] SP study in China, which found that providers increased antibiotic use from 55% to 85% when SPs requested antibiotics. Instead, we found that demanding an inappropriate antibiotic did not increase its use, but demanding an antiparasitic did in the Kenyan private sector.[7]

Our findings complement what was reported by Lopez et al,[46] who by comparing both provider and patient roles, assessed whether patients' demands influence overprescription and overuse of antimalarials in Mali.[46] With a large sample of real patients randomly assigned different information and malaria treatment subsidies across 60 health facilities in Bamako, the authors found that patients demanding resulted in higher rates of treatment than if providers were in control of dispensing vouchers. They additionally found that for more severe cases, providers were reluctant to provide inappropriate treatment, but that patient-driven demand resulted in an excess of treatment for milder cases.

Our findings also have implications on the literature on overdispensing of antimicrobial therapy and understanding quality of care outcomes that are related to antimicrobial resistance (AMR). Particularly, in Kenya there have been alarms raised about AMR for diarrhoeal infections and the government has launched a national AMR policy just before our study was implemented.[47 48] Although our study does not have enough data to address the AMR issue more deeply, we show how demanding an inappropriate medicine can result in higher rates of mismanagement of childhood illnesses than demanding other inappropriate medicines, which has implications for antimicrobial stewardship efforts, training on the consequences of overprescription, and quality improvement interventions. Our study is not able to speak to whether providers were given training on AMR but is able to shed light on how providers seem to be aware that certain medicines are inappropriate for most cases of childhood acute diarrhoea. We highlight what happens if a provider gives medicines demanded by a mother or caretaker of a child in the private sector where profits also matter.

## Possible explanations and implications

Providers may be trading off clinical benefits and risks with profits but doing so based on how concrete clinical consequences are with respect to what may be more appropriate for the presenting condition. A future study could examine this more in depth. Other factors likely play a role in determining prescribing practices, including the limited access to diagnostics to rule out conditions that do not require antimicrobial treatment. Further, we caution on extrapolation to other settings where knowledge and training may not be as high, since knowledge on other correct management outcomes appear to be higher in Kenya than in other LMIC settings for both infectious conditions as well as non-communicable diseases.[22 40 49]

In this study, we did not categorise the efficacy or safety of these drugs, since classifying the prescription of the medicines that were demanded in this study as 'harmful' may be misleading or lead to misinterpretation. The safety profile of both drugs in terms of side effects is very good, which is reassuring. Both amoxicillin and albendazole are well tolerated even in young children. However, this might provide the false perception of harmlessness which favours inappropriate use. In this specific context, the threats to public health likely are much greater than those to the individual. The inappropriate use of amoxicillin, though narrow-spectrum and less problematic than other antibiotics, could favour resistance selection among commensal and pathogenic bacteria. Similar considerations apply to albendazole, though the consequences of its widespread use are less studied especially in human medicine.

## CONCLUSION

In the setting of private primary care in Kenya, the SP method allowed us to assess providers with the same patient presentation and to causally infer the effects of patient characteristics and actions on quality of care. Most notably, we sought to investigate whether explicitly asking for amoxicillin (an unnecessary antibiotic) or albendazole (an unnecessary antiparasitic used for deworming) had an impact on correct case management and drug prescribing. We find that the provision of inappropriate medicines as one aspect of care quality can be influenced by patients demanding it, but depending on the drug, that may not always be the case. That providers increased the misuse of the antiparasitic deworming medicine but not the antibiotic amoxicillin suggests the need for future research on provider knowledge, awareness, and perceptions of profit and individual and community health trade-offs when making prescription decisions after patients demand specific inappropriate medicines.

**Acknowledgements** The authors would like to thank Nicole Perales, Rita Cuckovich, Afke Jagar and Innovations for Poverty Action Kenya for their technical and research assistance; to Jishnu Das and Amy Dolinger for their support; Stefano Bertozzi, Jonathan Kolstad, Aprajit Mahajan and the two reviewers for their comments and feedback; Catherine Wanjiru, John Wanangwe, Jacob Odipo, Daniel Waithaka for their advising; to Andrew Muriithi, Pheliciah Mwachofi, Purity Kimuru, Rodgers Kegode and Salome Omondi for their superb supervision; and to the SPs for their communication and hard work.

**Contributors** Conceptualisation: AK, CEB, GS, PJG. Funding acquisition: PJG, AK. Methodology: AK, PJG. Fieldwork supervision: AK. Coding of medicines and tests: AK, GS. Writing—original draft: AK, CEB, GS. Writing—review and editing: AK, GS, CEB, PJG. AK accepts full responsibility for the overall content as guarantor.

**Funding** This project was funded by a grant from the Bill and Melinda Gates Foundation (Opportunity ID#: OPP1044138). The former UK Department of International Development provided funding for the grant.

**Competing interests** None declared.

**Patient and public involvement** Patients and/or the public were not involved in the design, or conduct, or reporting, or dissemination plans of this research.

**Patient consent for publication** Not applicable.

**Ethics approval** This study involves human participants and was approved by Innovations for Poverty Action's IRB board (Protocol No. 1085) and Kenya Medical Research Institute (Non-SSC Protocol No. 372). The study also received local research permission in Kenya National Commission for Science, Technology and Innovation Permit #NA-COSTI/P/19/5343/28310. The following is an excerpt from our online supplemental appendix A.1.5 on ethical clearance: similar to other SP studies with similar designs and embedded in an intervention, we sought a waiver of provider informed consent to conduct the SP study. The request for a waiver was based on a recent study commissioned by the US Department of Health and Human Services to assess the ethics of simulated patient studies. Supported by a pilot study conducted in Nairobi that validated the SP method in the Kenyan context, both ethics committees approved the waiver request within the AHME evaluation study because (1) combining informed consent with the congregation of providers during trainings and the implementation of interventions during the study period posed threats to the scientific validity of the study objectives, as well as to the risk of SP detection, and (2) there is no more than minimal risk of participation to the SPs or providers, as reported in the Nairobi SP pilot and validation study (Daniels *et al*). Ethics committee approvals with the waiver of informed consent were provided conditional on our agreement to return to all clinics visited by SPs to disclose the SP study to them and to provide them with an opportunity to ask questions and discuss any concerns. During 1–23 January 2020, we informed all clinics and providers that received SPs and that were not closed permanently at that time. Online supplemental appendix A4 provides more details on ethical considerations for using the SP method for this study.

**Provenance and peer review** Not commissioned; externally peer reviewed.

**Data availability statement** Data are available in a public, open access repository. Individual deidentified interaction data with variables and code needed to recreate the results reported in this article are included and accessible at: https://github.com/kwantify/ahme_demanding/.

**ORCID iDs**
Ada Kwan http://orcid.org/0000-0003-4889-9433
Giorgia Sulis http://orcid.org/0000-0001-6641-0094

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
