## [Reviewer comments · BMJ Open]

ARTICLE DETAILS

TITLE (PROVISIONAL)	Do private providers give patients what they demand, even if it is inappropriate? A randomized study utilizing unannounced standardized patients in Kenya
AUTHORS	Kwan, Ada; Boone, Claire; Sulis, Giorgia; Gertler, Paul

VERSION 1 – REVIEW

REVIEWER	Sautmann, Anja World Bank Group, Development Economics Research Group
REVIEW RETURNED	12-Dec-2021

GENERAL COMMENTS	This paper compares SP visits to 200 private clinics in Kenya in which the tester demands a prescription of either amoxicillin or albendazole for a case of childhood diarrhea. The paper finds that relative demand for albendazole leads to a significant increase in prescription rates by 32 percentage points, whereas use of amoxicillin was not significantly affected. I enjoyed reading this paper and learning about aspects of treatment and prescription practices in Kenyan clinics. The description of the experiment is clear and the materials provide a comprehensive picture of the study procedures and results. All of my comments except (1) and (2) are expositional or requests for clarification and do not affect the reported results. It would be helpful to discuss potential limitations of the SP method for studying patient demand for specific drugs. (1) Effect of the AHME program: It would be helpful to see AHME treatment assignment shown in the balance table. Additionally, could the quality aspects of AHME influence prescription behavior and response to patient demand? If we would expect important differences, would it make sense to estimate the response of providers to patient demand separately for treated and untreated clinics? (2) Total number of medicines are reported but may include vitamins or supplements. Can the data speak to the incidence of polypharmacy, i.e., unambiguous treatment for two or more different underlying conditions (such as a parasitic and a bacterial infection)? (3) Standard errors and number of SP visits per clinic: please clarify how standard errors were computed. In places the text seems to suggest that several SP visited the same clinic (e.g. notes Table 3, p. 38 point 5 on SP sequencing). It is stated that
---

standard errors are clustered at the “clinic and individual standardized patient levels”. However, it appears that the reported results include only one observation per clinic.

(4) Description of the policy context: I read Figure A.1 with interest, as the national guidelines for diarrhea management seem to indicate awareness of the problem of antimicrobial overuse. It would help the reader put the experiment into context to learn more about the policy background in Kenya at the time of the intervention. How were current AMR policies disseminated and enforced? What did providers know about AMR, overall and in the treatment arms of AHME? I am not well acquainted with the situation in Kenya, but e.g. a report by the Global Antibiotic Resistance Partnership raises alarms about AMR for diarrheal infections (GARP 2011), and Kenya seemed to have a national AMR policy in place at the time of the study (Govt of Kenya, 2017). It would also be helpful to discuss to what extent the detailed policy context might influence the study results. AMR initiatives tend to focus on the most widely used drugs, so perhaps providers were more aware that they should resist patient demand when amoxicillin is demanded rather than albendazole. As a downside, it could be possible that the providers suspected an SP in one case but not the other.

(5) Standardized-patient method: the SP method has clear advantages related to the accuracy of measurement that the authors describe well in several places (pages 5, 7, appendix p32). The authors also mention limitations related to the types of conditions that can be tested on p12. However, there are other caveats discussed in the literature, such as the possibility of testers behaving in a way that confirms the study hypotheses (e.g. Currie, Lin & Zhang 2011, Currie, Lin & Meng 2014, Aujla et al. 2021). Discussions of audit studies on discrimination (Goldberg, 1996; Heckman, 1998) also point out that an observed supply response to specific auditor characteristics or trained behavior may not translate to differences in market outcomes. Applied to the SP context, we cannot know whether providers in general overprescribe antiparasitics but not antibiotics in response to patient demand, because we do not know how often real patients demand these drugs. Relatedly, we do not know what conclusions providers draw about patients who demand a drug if this is uncommon behavior (see also point on SP detection above). I believe the authors are referencing this issue in lines 36-40 on p12, but it could be more clearly discussed. Such a discussion does not take away from the significance of the finding that the SP simply naming albendazole can raise the rate at which it is prescribed by over 30 percentage points.

Minor comments:

- Personally I consider AMR an extremely important policy issue. However, I see a slight tension between the argument on p.5/6 that diarrhea is an important case because it is a major cause of mortality and morbidity, and the focus of the study on the over-treatment aspect of quality of care. The authors might consider reframing a little.

- Consider revising l.35-38 p.6 for clarity.

- Consider re-ordering 1st paragraph of "Data Sources" to define pre-demanding/post-demanding before stating observation counts.
- Table 2, last 4 rows: it appears to me that the order of magnitude for these is off (unless perhaps these numbers are per patient?).
- Table 3, notes: sentence "Single observations..." is unclear.
- Missing observations: please add a note on the missing observations, e.g. for "any antibiotic" and "any anti parasitic" (Tables 3, B1, B2).
- Typo on p.12, l.15: our study is about the overprescription and overuse of antimalarials in Mali.
- Appendix Figure A.1 has a couple of typos in the first box.

References:

Navneet Aujla, Yen-Fu Chen, Yasara Samarakoon, Anna Wilson, Natalia Grolmusová, Abimbola Ayorinde, Timothy P Hofer, Frances Griffiths, Celia Brown, Paramjit Gill, Christian Mallen, Jo Sartori, Richard J Lilford, Comparing the use of direct observation, standardized patients and exit interviews in low- and middle-income countries: a systematic review of methods of assessing quality of primary care, *Health Policy and Planning*, Volume 36, Issue 3, April 2021, Pages 341–356, <https://doi.org/10.1093/heapol/czaa152>.

Janet Currie, Wanchuan Lin, Juanjuan Meng, Addressing antibiotic abuse in China: An experimental audit study, *Journal of Development Economics*, Volume 110, 2014, Pages 39-51, <https://doi.org/10.1016/j.jdeveco.2014.05.006>.

Janet Currie, Wanchuan Lin, Wei Zhang, Patient knowledge and antibiotic abuse: Evidence from an audit study in China, *Journal of Health Economics*, Volume 30, Issue 5, 2011, Pages 933-949, <https://doi.org/10.1016/j.jhealeco.2011.05.009>.

Global Antibiotic Resistance Partnership—Kenya Working Group. 2011. *Situation Analysis and Recommendations: Antibiotic Use and Resistance in Kenya*. Washington, DC and New Delhi: Center for Disease Dynamics, Economics & Policy https://cddep.org/wp-content/uploads/2017/08/garp-kenya_sa.pdf

Goldberg, Pinelopi Koujianou, 1996. "Dealer Price Discrimination in New Car Purchases: Evidence from the Consumer Expenditure Survey" *Journal of Political Economy*, Volume 104, Number 3, <https://doi.org/10.1086/262035>

National Policy for the Prevention and Containment of Antimicrobial Resistance, Nairobi, Kenya: Government of Kenya, April 2017. https://www.health.go.ke/wp-content/uploads/2017/04/Kenya-AMR-Containment-Policy-Final_April.pdf

Heckman, James J. 1998. "Detecting Discrimination." *Journal of Economic Perspectives*, 12 (2): 101-116.

	DOI: 10.1257/jep.12.2.101
REVIEWER	Wang, Xiaohui Lanzhou University, School of Public Health
REVIEW RETURNED	20-Jan-2022

GENERAL COMMENTS	This study examined the role of patient demand for inappropriate care using the methods of USP and vignette data. Inappropriate care refers to the behavior of the providers. That is to say, whether the provider prescribes or dispenses the antibiotic amoxicillin or the deworming drug albendazole. This paper makes a valuable contribution to the study of quality of care using the SP method. Here are some Minor comments:  1. The format: please standardize the reference format in the text. (P10, Line 5 reference 23,23,34 vs. Line9 reference 21,23 2. Inconsistent page numbers throughout the text 3. Please follow the CONSORT reporting guideline to organize the manuscript. 4. Some comment regarding the main text are as follows: P2 Abstract Methods L19. According to the manuscript, I assume you used unannounced SP visits instead of SP visits? Would you please clarify? Results Line 28 of this study aims to examine the effect of quality of care when patients demand different types of inappropriate medicines. But according to the result, the outcome is dispensing rate of the two types of medicine. Is there any possibility to measure the quality of care more specifically? Is there any underlining incentive-induced difference if the patient requires the medicine at a different time point? P6 Line 35. is the number (36) typo? Or does this refer to reference 36? Methods Data Sources P37 Line 22. The SPs were locally recruited and trained in January 2019. The last day of the training was 01-Feb-19, while the visits were conducted from March 8 to May 28. One concern is that after two months, the SPs might forget some of the scripts? Any quality control before the fieldwork? P37 Line 24. This study described three time points when the SP was assigned to the “demanding dawa ya minyao” group. Will the first one affect the providers' behavior? Say, change the prescribe itself? P37 Line 39 The same concern. P8 Line 8. It is a good way to conduct the provider survey among those who saw the SPs to explore the know-do gap. Outcomes P8 Line 14. What is the difference between prescribe and dispense? Results P9 Line 27. why measure the person waiting in the waiting room when the SP arrived? Discussion P10 Line 16-30. the authors described the result of anecdotal narratives during debriefs with the supervisors of the SP fieldwork.
---

	It might be easier to understand the method used in this study. If I were the author, I would like reorganize this section. To be specific, move these to “results” as part of the qualitative research method. P19. I failed to find Figure 1. Clinic sample and SP randomized study design. P22. In Table 3. Summary statistics of SP visits, under “Visit characteristics”, it seems that there are some indicators with missing data. e.g. “Minutes spent with provider”, just wondering, is this due to SP failed to collect this information? Or other reason?
--	---

VERSION 1 – AUTHOR RESPONSE

Reviewer 1 comments and author responses:

Dr. Anja Sautmann, World Bank Group

Comments to the Author:

This paper compares SP visits to 200 private clinics in Kenya in which the tester demands a prescription of either amoxicillin or albendazole for a case of childhood diarrhea. The paper finds that relative demand for albendazole leads to a significant increase in prescription rates by 32 percentage points, whereas use of amoxicillin was not significantly affected.

I enjoyed reading this paper and learning about aspects of treatment and prescription practices in Kenyan clinics. The description of the experiment is clear and the materials provide a comprehensive picture of the study procedures and results.

All of my comments except (1) and (2) are expositional or requests for clarification and do not affect the reported results. It would be helpful to discuss potential limitations of the SP method for studying patient demand for specific drugs.

We are appreciative that you took time to review our manuscript. We would like to kindly note that our quantitative results changed slightly based on incorporating the helpful feedback and comments from you and the other reviewer. Changes were made specifically to: variables to assess balance in Table 1, clustering of standard errors, missing data, and the addition of a polypharmacy outcome. Overall, reported findings retain the same conclusions. Your comments and questions are addressed point-by-point below.

(1) Effect of the AHME program: It would be helpful to see AHME treatment assignment shown in the balance table. Additionally, could the quality aspects of AHME influence prescription behavior and response to patient demand? If we would expect important differences, would it make sense to estimate the response of providers to patient demand separately for treated and untreated clinics?

Thank you for this comment and raising these important questions. We respond to each of your points below in turn.

First, we agree that it is helpful to show balance for the AHME treatment assignment. We tested if there was any AHME treatment difference between the clinics assigned to receive an SP demanding albendazole or amoxicillin. We find in the analytic sample that 60% of the 102 clinics assigned albendazole and 46% of the 98 clinics assigned amoxicillin were also assigned to receive the AHME program (p-value = 0.05). We continue to include an AHME treatment indicator (1=assigned AHME treatment; 0, otherwise) in all our analyses to control for the effects of the independently randomized program. We are hesitant to include this variable in the main balance table as this would warrant a longer description of the program and the inclusion of the AHME treatment variable in the main tables (we will get back to this in a moment). We include the following text and Appendix Table A1 (p. S5):

The SP experiments were randomly assigned independent of the AHME treatment assignment. The table below shows the balance of AHME assignment across the SP demanding experiment assignments for our analytic sample. We include a AHME treatment indicator for analyses based on the clinic assignment to the AHME treatment or control group.

	Clinics Assigned to Receive an SP Demanding Albendazole, n = 102			Clinics Assigned to Receive an SP Demanding Amoxicillin, n = 98			p-value
	N	Mean	95% CI	N	Mean	95% CI	
Randomly assigned to receive AHME program	102	0.60	(0.50 – 0.69)	98	0.46	(0.36 – 0.56)	0.050

Second, the reviewer also raises very important considerations regarding the AHME program and the random assignment of clinics to the program. We also asked ourselves whether quality aspects of AHME influenced prescription behavior and response to patient demand while writing this manuscript. Based on the reviewer’s comments, we present the following tables to show the effect of the AHME program in the same analyses we present the demanding experiment (“AHME treatment” indicator) without and with interactions:

- (a) N=400 observations (pre-demanding and post-demanding) without AHME and Demanding interactions

	(1)	(2)	(3)	(4)	(5)	(6)	(7)	(8)	(9)
	Correct case management	Asked to return or referred	Asked to return	Referred elsewhere	Gave or advised on ORS	Advised on ORS	Dispensed/prescribed: ORS	Dispensed/prescribed: Zinc	Number of medicines
Albendazole post-demanding									
Coefficient	0.117	0.110	0.123	0.014	0.053	0.198	-0.035	-0.060	0.010
Standard Error	(0.036)	(0.038)	(0.038)	(0.015)	(0.038)	(0.042)	(0.032)	(0.032)	(0.134)
p-value	[0.001]	[0.004]	[0.001]	[0.352]	[0.166]	[0.000]	[0.279]	[0.063]	[0.939]
Amoxicillin post-demanding									
Coefficient	0.062	0.090	0.076	-0.015	0.108	0.171	0.037	0.063	-0.011
Standard Error	(0.038)	(0.040)	(0.040)	(0.016)	(0.041)	(0.047)	(0.034)	(0.034)	(0.140)
p-value	[0.101]	[0.026]	[0.060]	[0.351]	[0.009]	[0.000]	[0.281]	[0.065]	[0.939]
AHME treatment									
Coefficient	-0.009	0.024	0.093	-0.001	-0.069	-0.167	0.085	0.007	0.027
Standard Error	(0.064)	(0.056)	(0.057)	(0.033)	(0.069)	(0.061)	(0.069)	(0.071)	(0.278)
p-value	[0.886]	[0.669]	[0.105]	[0.985]	[0.313]	[0.006]	[0.221]	[0.923]	[0.923]
Observations	400	400	400	400	400	400	400	400	400
Pre-demanding Group Mean	0.705	0.245	0.255	0.060	0.570	0.448	0.360	0.390	2.375
	(10)	(11)	(12)	(13)	(14)	(15)	(16)	(17)	
	Number of efficacious medicines	Number of non-efficacious medicines	Any non-efficacious medicines	Dispensed/prescribed: Albendazole	Dispensed/prescribed: Amoxicillin	Dispensed/prescribed: Antibiotics	Dispensed/prescribed: Antiparasitics	Dispensed/prescribed: Antibiotics & Antiparasitics	
Albendazole post-demanding									
Coefficient	-0.096	0.106	-0.015	0.142	-0.017	0.008	0.122	0.094	
Standard Error	(0.060)	(0.105)	(0.033)	(0.027)	(0.019)	(0.034)	(0.029)	(0.027)	
p-value	[0.112]	[0.315]	[0.638]	[0.000]	[0.383]	[0.810]	[0.000]	[0.001]	
Amoxicillin post-demanding									
Coefficient	0.100	-0.110	0.016	-0.148	0.018	-0.009	-0.127	-0.098	
Standard Error	(0.063)	(0.109)	(0.034)	(0.027)	(0.020)	(0.036)	(0.029)	(0.028)	
p-value	[0.114]	[0.313]	[0.640]	[0.000]	[0.383]	[0.810]	[0.000]	[0.000]	
AHME treatment									
Coefficient	0.092	-0.065	-0.104	-0.003	0.004	-0.030	-0.056	-0.058	
Standard Error	(0.127)	(0.217)	(0.068)	(0.054)	(0.040)	(0.072)	(0.061)	(0.057)	
p-value	[0.471]	[0.765]	[0.127]	[0.959]	[0.913]	[0.680]	[0.353]	[0.314]	
Observations	400	400	400	400	400	400	400	400	
Pre-demanding Group Mean	0.750	1.625	0.670	0.190	0.095	0.555	0.245	0.205	

Note: The table shows ordinary least squares regressions using standardized patient (SP) data. Robust standard errors are in parentheses, clustered at the clinic level (2 observations corresponding to 1 SP visit per clinic). Two-sided p-values in brackets. All models contain SP fixed effects and control for the 0-1 AHME treatment indicator, a binary indicator for whether a clinic was assigned to receive an SP demanding albendazole at the end of the visit (Albendazole post-demanding) or whether a clinic was assigned to receive an SP demanding amoxicillin at the end of the visit (Amoxicillin post-demanding). All outcomes in models (1)-(17) are binary variables where if the action occurred during the visit 1=yes; 0=otherwise for both pre-demanding and post-demanding time points for the visit. Correct case management is a binary outcome for whether any one of the following actions were performed according to guidelines: asked to return, referred elsewhere, gave ORS, or advised on ORS. ORS is oral rehydration salts. Antiparasitics include antimalarials. "Dispensed/prescribed: Antibiotics & Antiparasitics" refers to whether the provider gave any antibiotic and any antiparasitic.

(b) N=400 observations (pre-demanding and post-demanding) with AHME and Demanding interactions

	(1)	(2)	(3)	(4)	(5)	(6)	(7)	(8)	(9)
	Correct case management	Asked to return or referred	Asked to return	Referred elsewhere	Gave or advised on ORS	Advised on ORS	Dispensed/prescribed: ORS	Dispensed/prescribed: Zinc	Number of medicines
Albendazole post-demanding									
Coefficient	0.132	0.098	0.112	0.014	0.092	0.218	-0.059	-0.110	-0.254
Standard Error	(0.059)	(0.059)	(0.057)	(0.027)	(0.063)	(0.063)	(0.056)	(0.059)	(0.214)
p-value	[0.027]	[0.100]	[0.053]	[0.592]	[0.144]	[0.001]	[0.299]	[0.063]	[0.237]
Albendazole * AHME treatment									
Coefficient	-0.027	0.023	0.020	-0.001	-0.067	-0.033	0.040	0.084	0.444
Standard Error	(0.073)	(0.077)	(0.077)	(0.032)	(0.077)	(0.086)	(0.070)	(0.071)	(0.275)
p-value	[0.718]	[0.768]	[0.794]	[0.981]	[0.384]	[0.700]	[0.569]	[0.242]	[0.109]
Amoxicillin post-demanding									
Coefficient	0.067	0.057	0.046	-0.011	0.099	0.171	0.045	0.085	0.196
Standard Error	(0.052)	(0.049)	(0.048)	(0.021)	(0.055)	(0.065)	(0.043)	(0.045)	(0.165)
p-value	[0.195]	[0.249]	[0.346]	[0.591]	[0.073]	[0.009]	[0.298]	[0.062]	[0.236]
Amoxicillin * AHME treatment									
Coefficient	-0.011	0.069	0.065	-0.007	0.023	0.001	-0.020	-0.050	-0.454
Standard Error	(0.075)	(0.079)	(0.081)	(0.032)	(0.081)	(0.091)	(0.070)	(0.070)	(0.286)
p-value	[0.884]	[0.380]	[0.424]	[0.817]	[0.776]	[0.993]	[0.777]	[0.481]	[0.115]
AHME treatment									
Coefficient	0.000	0.001	0.071	0.001	-0.059	-0.159	0.080	-0.001	0.035
Standard Error	(0.070)	(0.056)	(0.056)	(0.033)	(0.073)	(0.065)	(0.069)	(0.071)	(0.277)
p-value	[0.999]	[0.986]	[0.206]	[0.965]	[0.421]	[0.014]	[0.246]	[0.991]	(0.277)
Observations	400	400	400	400	400	400	400	400	400
Pre-demanding Group Mean	0.705	0.245	0.255	0.060	0.570	0.448	0.360	0.390	2.375
	(10)	(11)	(12)	(13)	(14)	(15)	(16)	(17)	
	Number of efficacious medicines	Number of non-efficacious medicines	Any non-efficacious medicines	Dispensed/prescribed: Albendazole	Dispensed/prescribed: Amoxicillin	Dispensed/prescribed: Antibiotics	Dispensed/prescribed: Antiparasitics	Dispensed/prescribed: Antibiotics & Antiparasitics	
Albendazole post-demanding									
Coefficient	-0.169	-0.085	-0.094	0.176	-0.030	-0.057	0.143	0.118	
Standard Error	(0.105)	(0.165)	(0.053)	(0.049)	(0.030)	(0.058)	(0.053)	(0.050)	
p-value	[0.111]	[0.607]	[0.080]	[0.000]	[0.321]	[0.329]	[0.008]	[0.019]	
Albendazole * AHME treatment									
Coefficient	0.124	0.320	0.132	-0.059	0.022	0.110	-0.037	-0.042	
Standard Error	(0.129)	(0.216)	(0.067)	(0.058)	(0.039)	(0.072)	(0.063)	(0.060)	
p-value	[0.339]	[0.140]	[0.050]	[0.311]	[0.579]	[0.129]	[0.559]	[0.479]	
Amoxicillin post-demanding									
Coefficient	0.130	0.066	0.072	-0.136	0.023	0.044	-0.111	-0.091	
Standard Error	(0.081)	(0.128)	(0.041)	(0.040)	(0.023)	(0.045)	(0.042)	(0.039)	
p-value	[0.109]	[0.607]	[0.081]	[0.001]	[0.323]	[0.329]	[0.009]	[0.021]	
Amoxicillin * AHME treatment									
Coefficient	-0.070	-0.384	-0.124	-0.023	-0.012	-0.115	-0.033	-0.012	
Standard Error	(0.129)	(0.225)	(0.069)	(0.056)	(0.041)	(0.072)	(0.061)	(0.057)	
p-value	[0.589]	[0.090]	[0.072]	[0.684]	[0.770]	[0.112]	[0.585]	[0.841]	
AHME treatment									
Coefficient	0.080	-0.044	-0.104	0.017	0.002	-0.027	-0.039	-0.044	
Standard Error	(0.127)	(0.217)	(0.067)	(0.057)	(0.040)	(0.072)	(0.063)	(0.059)	
p-value	[0.531]	[0.839]	[0.124]	[0.764]	[0.955]	[0.709]	[0.538]	[0.451]	
Observations	400	400	400	400	400	400	400	400	
Pre-demanding Group Mean	0.750	1.625	0.670	0.190	0.095	0.555	0.245	0.205	

Note: The table shows ordinary least squares regressions using standardized patient (SP) data. Robust standard errors are in parentheses, clustered at the clinic level (2 observations corresponding to 1 SP visit per clinic). Two-sided p-values in brackets. All models contain SP fixed effects and control for the 0-1 AHME treatment indicator, a binary indicator for whether a clinic was assigned to receive an SP demanding albendazole at the end of the visit (Albendazole post-demanding) or whether a clinic was assigned to receive an SP demanding amoxicillin at the end of the visit (Amoxicillin post-demanding). Models also include interactions between the AHME treatment and each of the demanding experiments. All outcomes in models (1)-(17) are binary variables where if the action occurred during the visit 1=yes; 0=otherwise for both pre-demanding and post-demanding time points for the visit. Correct case management is a binary outcome for whether any one of the following actions were performed according to guidelines: asked to return, referred elsewhere, gave ORS, or advised on ORS. ORS is oral rehydration salts. Antiparasitics include antimalarials. "Dispensed/prescribed: Antibiotics & Antiparasitics" refers to whether the provider gave any antibiotic and any antiparasitic.

(c) N=200 observations (post-demanding only) without AHME and Demanding interactions

	(1)	(2)	(3)	(4)	(5)	(6)	(7)	(8)	(9)
	Correct case management	Asked to return or referred	Asked to return	Referred elsewhere	Gave or advised on ORS	Advised on ORS	Dispensed/prescribed: ORS	Dispensed/prescribed: Zinc	Number medicines
Albendazole post-demanding									
Coefficient	0.052	0.014	0.046	0.034	-0.074	-0.015	-0.086	-0.148	0.025
Standard Error	(0.073)	(0.076)	(0.077)	(0.040)	(0.081)	(0.080)	(0.081)	(0.081)	(0.331)
p-value	[0.473]	[0.853]	[0.553]	[0.391]	[0.360]	[0.854]	[0.287]	[0.071]	[0.940]
AHME treatment									
Coefficient	-0.022	0.049	0.115	-0.003	-0.083	-0.168	0.092	0.018	0.025
Standard Error	(0.064)	(0.067)	(0.068)	(0.035)	(0.071)	(0.071)	(0.071)	(0.072)	(0.292)
p-value	[0.727]	[0.466]	[0.091]	[0.926]	[0.248]	[0.019]	[0.200]	[0.800]	[0.932]
Observations	200	200	200	200	200	200	200	200	200
Demanding Amoxicillin Group Mean	0.755	0.296	0.286	0.051	0.673	0.582	0.398	0.449	2.347
	(10)	(11)	(12)	(13)	(14)	(15)	(16)	(17)	
	Number of efficacious medicines	Number of non-efficacious medicines	Any non-efficacious medicines	Dispensed/prescribed: Albendazole	Dispensed/prescribed: Amoxicillin	Dispensed/prescribed: Antibiotics	Dispensed/prescribed: Antiparasitics	Dispensed/prescribed: Antibiotics & Antiparasitics	
Albendazole post-demanding									
Coefficient	-0.234	0.259	-0.038	0.348	-0.041	0.020	0.299	0.230	
Standard Error	(0.147)	(0.257)	(0.078)	(0.059)	(0.049)	(0.084)	(0.068)	(0.066)	
p-value	[0.113]	[0.314]	[0.631]	[0.000]	[0.397]	[0.810]	[0.000]	[0.001]	
AHME treatment									
Coefficient	0.110	-0.085	-0.101	-0.030	0.008	-0.031	-0.080	-0.075	
Standard Error	(0.130)	(0.226)	(0.069)	(0.052)	(0.043)	(0.074)	(0.060)	(0.058)	
p-value	[0.398]	[0.708]	[0.147]	[0.570]	[0.859]	[0.671]	[0.188]	[0.197]	
Observations	200	200	200	200	200	200	200	200	
Demanding Amoxicillin Group Mean	0.847	1.500	0.704	0.031	0.112	0.551	0.133	0.122	

Note: The table shows ordinary least squares regressions using standardized patient (SP) data for the post-demanding phase of the N=200 SP visits (1 observation corresponds to 1 SP visit per clinic). Standard errors are in parentheses. Two-sided p-values in brackets. All models contain SP fixed effects and control for the 0-1 AHME treatment indicator, a binary indicator for whether the visit was Albendazole post-demanding (if 0, the visit was Amoxicillin post-demanding). All outcomes in models (1)-(17) are binary variables where if the action occurred during by the end of the visit 1=yes; 0=otherwise. Correct case management is a binary outcome for whether any one of the following actions were performed according to guidelines: asked to return, referred elsewhere, given ORS, or advised on ORS. ORS is oral rehydration salts. Antiparasitics include antimalarials. "Dispensed/prescribed: Antibiotics & Antiparasitics" refers to whether the provider gave any antibiotic and any antiparasitic.

(d) N=200 observations (pre-demanding and post-demanding) with AHME and Demanding interactions

	(1)	(2)	(3)	(4)	(5)	(6)	(7)	(8)	(9)
	Correct case management	Asked to return or referred	Asked to return	Referred elsewhere	Gave or advised on ORS	Advised on ORS	Dispensed/prescribed: ORS	Dispensed/prescribed: Zinc	Number medicines
Albendazole post-demanding									
Coefficient	0.064	0.033	0.064	0.031	-0.022	0.006	-0.117	-0.217	-0.450
Standard Error	(0.099)	(0.102)	(0.104)	(0.054)	(0.109)	(0.109)	(0.109)	(0.110)	(0.445)
p-value	[0.519]	[0.745]	[0.541]	[0.572]	[0.840]	[0.954]	[0.284]	[0.049]	[0.313]
Albendazole * AHME treatment									
Coefficient	-0.022	-0.037	-0.034	0.007	-0.100	-0.040	0.060	0.134	0.913
Standard Error	(0.127)	(0.132)	(0.134)	(0.070)	(0.141)	(0.140)	(0.141)	(0.142)	(0.574)
p-value	[0.865]	[0.778]	[0.797]	[0.918]	[0.478]	[0.774]	[0.672]	[0.347]	[0.113]
AHME treatment									
Coefficient	-0.012	0.067	0.132	-0.007	-0.034	-0.148	0.062	-0.047	-0.420
Standard Error	(0.089)	(0.093)	(0.094)	(0.049)	(0.099)	(0.099)	(0.099)	(0.100)	(0.404)
p-value	[0.894]	[0.472]	[0.164]	[0.890]	[0.733]	[0.135]	[0.528]	[0.639]	[0.300]
Observations	200	200	200	200	200	200	200	200	200
Demanding Amoxicillin Group Mean	0.755	0.296	0.286	0.051	0.673	0.582	0.398	0.449	2.347
	(10)	(11)	(12)	(13)	(14)	(15)	(16)	(17)	
	Number of efficacious medicines	Number of non-efficacious medicines	Any non-efficacious medicines	Dispensed/prescribed: Albendazole	Dispensed/prescribed: Amoxicillin	Dispensed/prescribed: Antibiotics	Dispensed/prescribed: Antiparasitics	Dispensed/prescribed: Antibiotics & Antiparasitics	
Albendazole post-demanding									
Coefficient	-0.335	-0.115	-0.173	0.365	-0.059	-0.099	0.299	0.244	
Standard Error	(0.198)	(0.345)	(0.105)	(0.080)	(0.066)	(0.113)	(0.092)	(0.089)	
p-value	[0.094]	[0.738]	[0.101]	[0.000]	[0.373]	[0.380]	[0.001]	[0.007]	
Albendazole * AHME treatment									
Coefficient	0.193	0.720	0.260	-0.031	0.034	0.229	0.001	-0.028	
Standard Error	(0.256)	(0.444)	(0.135)	(0.104)	(0.085)	(0.145)	(0.119)	(0.115)	
p-value	[0.451]	[0.107]	[0.056]	[0.763]	[0.693]	[0.116]	[0.994]	[0.808]	
AHME treatment									
Coefficient	0.016	-0.436	-0.227	-0.015	-0.009	-0.143	-0.080	-0.062	
Standard Error	(0.180)	(0.313)	(0.095)	(0.073)	(0.060)	(0.102)	(0.084)	(0.081)	
p-value	[0.931]	[0.165]	[0.018]	[0.842]	[0.884]	[0.163]	[0.341]	[0.446]	
Observations	200	200	200	200	200	200	200	200	
Demanding Amoxicillin Group Mean	0.847	1.500	0.704	0.031	0.112	0.551	0.133	0.122	

Note: The table shows ordinary least squares regressions using standardized patient (SP) data for the post-demanding phase of the N=200 SP visits (1 observation corresponds to 1 SP visit per clinic). Standard errors are in parentheses. Two-sided p-values in brackets. All models contain SP fixed effects and control for the 0-1 AHME treatment indicator, a binary indicator for whether the visit was Albendazole post-demanding (if 0, the visit was Amoxicillin post-demanding). Models also include interactions between the AHME treatment and the Albendazole post-demanding experiment. All outcomes in models (1)-(17) are binary variables where if the action occurred during by the end of the visit 1=yes; 0=otherwise. Correct case management is a binary outcome for whether any one of the following actions were performed according to guidelines: asked to return, referred elsewhere, gave ORS, or advised on ORS. ORS is oral rehydration salts. Antiparasitics include antimalarials. "Dispensed/prescribed: Antibiotics & Antiparasitics" refers to whether the provider gave any antibiotic and any antiparasitic.

The above tables (a) and (b) are now Appendix Table B1 (a) and (b) on pp. S17-S18.

The above tables (c) and (d) are now Appendix Table B2 (a) and (b) on pp. S19-S20.

(Also note: The analyses have been updated to correspond to the reviewer’s comment (2) related to polypharmacy and (3) related to clustering – see below.)

After much thought, we elected to not report or show the effects of demanding separately for AHME treated and untreated clinics in the main text for three reasons: (1) quality aspects of AHME do not influence prescription behavior and response to patient demand, (2) the program did not directly intend to influence provider prescribing behavior, and (3) the quality aspects as they relate to the program are outside the scope of this study. Because of this, we refer to the balance results in the main text and refer the reader to Appendix Table A1, B1, and B2. We present a more thorough discussion of the effects of the AHME program on quality in a separate manuscript in preparation.

(2) Total number of medicines are reported but may include vitamins or supplements. Can the data speak to the incidence of polypharmacy, i.e., unambiguous treatment for two or more different underlying conditions (such as a parasitic and a bacterial infection)?

We thank the reviewer for this insightful comment and question. We do feel that we can better address the unambiguous treatment specifically for both a parasitic and a bacterial infection. However, since the average number of medicines given out were 2.38 (pooled SP visits), we report a dichotomous variable for whether the visit resulted in any antibiotic and any antiparasitic, including antimalarials.

Specifically, in Table 3 (p. 22), we add the variable “Dispensed/ prescribed: antibiotics and antiparasitics”, which refers to whether the provider gave any antibiotic and any antiparasitic (defined in the table notes). We also included this outcome in Figure 2 and Appendix Tables B1 and B2.

	Pooled SP Visits, n = 200		SP Visits Demanding Albendazole, n = 102		SP Visits Demanding Amoxicillin, n = 98		(3)-(2) difference in means t-test
	N	Mean	N	Mean	N	Mean	p-value
Dispensed/ prescribed: Antiparasitics	200	0.25	102	0.35	98	0.13	0.000
Dispensed/ prescribed: Antibiotics	200	0.56	102	0.56	98	0.55	0.912
Dispensed/ prescribed: Antibiotics & Antiparasitics	200	0.21	102	0.28	98	0.12	0.004

(3) Standard errors and number of SP visits per clinic: please clarify how standard errors were computed. In places the text seems to suggest that several SP visited the same clinic (e.g. notes Table 3, p. 38 point 5 on SP sequencing). It is stated that standard errors are clustered at the “clinic and individual standardized patient levels”. However, it appears that the reported results include only one observation per clinic.

We thank the reviewer for bringing this to our attention. The reviewer is correct in that we have one visit conducted by one SP per clinic. We have considered this carefully and following Abadi, Athey, Imbens, and Wooldridge (2017) “When should you adjust standard errors for clustering?”, we aimed to cluster standard errors at the level of SP demanding treatment assignment. In this case, the treatment (or independent variable) is the assignment of an SP to a clinic. Therefore, in analyses where we include one SP observation per clinic, we do not cluster standard errors (Figure 2 and Appendix Table B2; where N=200 observations). In analyses where we include 2 observations per clinic, we clustered standard errors at the clinic level (see Appendix table B1; where N=400 observations).

We read through the text carefully to make sure we have removed any places in the text that: (i) describe clustered standard errors except in Appendix Table B1; and (ii) suggest more than one SP ever visited a single clinic (e.g., notes in Table 3 and Figure 2).

(4) Description of the policy context: I read Figure A.1 with interest, as the national guidelines for diarrhea management seem to indicate awareness of the problem of antimicrobial overuse. It would help the reader put the experiment into context to learn more about the policy background in Kenya at the time of the intervention.

How were current AMR policies disseminated and enforced? What did providers know about AMR, overall and in the treatment arms of AHME? I am not well acquainted with the situation in Kenya, but e.g. a report by the Global Antibiotic Resistance Partnership raises alarms about AMR for diarrheal infections (GARP 2011), and Kenya seemed to have a national AMR policy in place at the time of the study (Govt of Kenya, 2017).

It would also be helpful to discuss to what extent the detailed policy context might influence the study results. AMR initiatives tend to focus on the most widely used drugs, so perhaps providers were more aware that they should resist patient demand when amoxicillin is demanded rather than albendazole. As a downside, it could be possible that the providers suspected an SP in one case but not the other.

We are very appreciative of this comment and thank you for sharing references to the GARP report and the GoK AMR policy. We agree that antimicrobial overuse is a very important problem, and in fact we discussed this before submission amongst ourselves for the reasons you put forth.

Given the tension of this, as you point out, we discussed further. We have cited the GARP report and AMR policy and describe how our study is not able to speak to whether providers were given or exposed to training on AMR (references 46 and 47).

Further, with our data, we do not find ourselves in the position to respond to how AMR policies were disseminated and enforced across the country. AMR policies were also outside the scope of the AHME program aims. We reviewed the current national policy document issued by the Government of Kenya and it mainly provides high level guidance for reducing the burden of antibiotic resistance. Since this study was implemented only shortly after the release of the policy document, we feel not enough time has passed to assess change related to the policy.

We can stress the fact that there is a know-do gap and feel like our study can shed light on this through providing information on the actual actions of providers when faced with a SP demanding inappropriate medications. Providers seem to be aware that amoxicillin is not appropriate for most cases of acute diarrhea enough to not give it.

Given all of these points, we have worked to address the reviewer's important comment with the following text to the discussion (p. 11 line 39 – p. 12 line 17 see below)

Our findings also have implications on the literature on overdispensing of antimicrobial therapy and understanding quality of care outcomes that are related to antimicrobial resistance (AMR). Particularly, in Kenya there have been alarms raised about AMR for diarrheal infections and the Government has launched a national AMR policy just before our study was implemented.(46,47) Although our study does not have enough data to address the AMR issue more deeply, we show how demanding an inappropriate medicine can result in higher rates of mismanagement of childhood illnesses than demanding other inappropriate medicines, which has implications for antimicrobial stewardship efforts, training on the consequences of overprescription, and quality improvement interventions. Our study is not able to speak to whether providers were given training on AMR but is able to shed light on how providers seem to be aware that certain medicines are inappropriate for most cases of childhood acute diarrhea. We highlight what happens if a provider gives medicines demanded by a mother or caretaker of a child.

(5) Standardized-patient method: the SP method has clear advantages related to the accuracy of measurement that the authors describe well in several places (pages 5, 7, appendix p32). The authors also mention limitations related to the types of conditions that can be tested on p12. However, there are other caveats discussed in the literature, such as the possibility of testers behaving in a way that confirms the study hypotheses (e.g. Currie, Lin & Zhang 2011, Currie, Lin & Meng 2014, Aujla et al. 2021).

Thank you for this comment. We agree and have worked to make this clearer. We have added the following text to the Results section (p. 10, lines 33-37):

It is important to recognize that SPs are not real patients and can behave in a way that confirms the study hypotheses which has been discussed in previous SP studies.(6,7,41) However, if this were the case for our study, the effects would likely be non-differential with

respect to the type of medicine being demanded. Instead, an increased rate is only observed after demanding albendazole, and not after demanding amoxicillin.

Discussions of audit studies on discrimination (Goldberg, 1996; Heckman, 1998) also point out that an observed supply response to specific auditor characteristics or trained behavior may not translate to differences in market outcomes. Applied to the SP context, we cannot know whether providers in general overprescribe antiparasitics but not antibiotics in response to patient demand, because we do not know how often real patients demand these drugs.

We appreciate this link to the literature on the use of audit studies on discrimination. The reviewer raises an important discussion point. We have added the references suggested by the reviewer and have added the following text to the Results section (p. 10, line 38 – p. 11, line 12):

Third, given that we only examine the interaction between providers and SPs, we do not report on the role of care-seeking behavior and thus interpret findings conditional on patients seeking care. Further, since SPs are not real patients, what was found with SPs may not exactly reflect what happens with real patients, nor are we able to report on how satisfied real patients would have been given these prescription patterns. Similar to discussions in audit studies on discrimination, provider behaviors captured in this study as a response to SP features or trained characteristics may not translate to actual practice behaviors with real patients.(42,43) Our study does not conduct a detection survey to measure the extent of provider suspicion, but other SP studies with detection surveys find very low detection rates (0-5%).(8,23,44) In the study where we based our childhood diarrhea SP case, Daniels et al. (2017) administered a structured questionnaire two weeks after the completion of SP fieldwork in Nairobi, Kenya and found that despite providers having detected SPs in 9 instances, none of these actually matched the study's SP visits. As described earlier, what the SP method allows us to do which other methods cannot is to identify what happens across multiple providers when providers are presented with SPs randomly assigned to demand different inappropriate medicines, with the same presentation otherwise.

Relatedly, we do not know what conclusions providers draw about patients who demand a drug if this is uncommon behavior (see also point on SP detection above). I believe the authors are referencing this issue in lines 36-40 on p12, but it could be more clearly discussed. Such a discussion does not take away from the significance of the finding that the SP simply naming albendazole can raise the rate at which it is prescribed by over 30 percentage points.

Thank you for this comment. We agree and worked to strengthen the limitations and discussion. We note that multiple SPs visited different providers, and no providers raised suspicion. We hope we have adequately addressed this by including the following text to Appendix A3 (pp. S10-S11):

We developed and finalized the SP script and demanding experiment together with a group of five field supervisors from Kenya, 40 individuals from Kenya who were recruited and hired to be standardized patients for this study (and approximately 60 more who were recruited and

underwent partial training but not hired), and a technical advisory group of 4 health care providers who at the time of the study advised on national guidelines and actively trained cadres of health care providers. All of these individuals played a role in days of discussions and exercises during training on what medicines were trusted in the community and whether people in the community are open to using them. SPs and supervisors were involved in piloting the demanding of inappropriate medicines in the field. The team together acknowledged that amoxicillin and albendazole were common medicines, and their selection for study was not done arbitrarily. Further, we conducted the SP pilot with SPs demanding these two medicines before the actual study. The selection of these two medicines in the script above were the result of the training and piloting process.

From the experience before fieldwork for this study, the SP recruits, supervisors, and our technical advisory group did not find that it was uncommon for patients in Kenya to ask for specific medicines they are familiar with. In particular, amoxicillin and albendazole are commonly prescribed drugs in the study setting, and thus presumed patients demanding either of those would not be seen as suspicious. It should be noted that the SP scripts were developed while taking into account local habits and behaviors in order to minimize the risk of SPs being identified as simulated, standardized patients.

We do want to emphasize that SPs are not a perfect methodology, so what was found with SPs doesn't fully reflect what happens with real patients, as the reviewer mentions. We have taken additional care to make sure that this is clear (see excerpts above).

Minor comments:

- Personally I consider AMR an extremely important policy issue. However, I see a slight tension between the argument on p.5/6 that diarrhea is an important case because it is a major cause of mortality and morbidity, and the focus of the study on the over-treatment aspect of quality of care. The authors might consider reframing a little.

We agree and refer the reviewer to our comments to our responses to comment (4) above. We want to also emphasize again that we do think that both childhood illnesses and AMR issues in public health are important. We don't think there is enough data to address the AMR issue more deeply, but we do want to link the mismanagement of childhood illnesses to the implications to AMR and underline the consequences of overprescription, issues that warrant additional research. We have added on p. 11 line 43 – p. 12 line 4:

Although our study does not have enough data to address the AMR issue more deeply, we show how demanding an inappropriate medicine can result in higher rates of mismanagement of childhood illnesses than demanding other inappropriate medicines, which has implications for antimicrobial stewardship efforts, training on the consequences of overprescription, and quality improvement interventions.

- Consider revising l.35-38 p.6 for clarity.

Thank you. We have revised these lines as such (p. 6 line 43 – p. 7 line 3):

The SP method has the advantage that the researchers know the true condition of the 'patient' which is not possible when examining data derived from real patients. SP data particularly allows for providers across different facilities to be compared against the exact same patient scenario and is thus increasingly considered the gold standard for measuring provider practice across a sample of providers that lack standardized health records.

- Consider re-ordering 1st paragraph of "Data Sources" to define pre-demanding/post-demanding before stating observation counts.

Thank you for this suggestion. We agree with the reviewer that this reorganization is clearer. The text now reads as such (with the underlined text reflecting the sentence that previously later in the paragraph but now has been moved; p. 6, lines 14-17):

Between March 8 and May 28, 2019, 200 successful SP visits were conducted at 200 private Kenyan clinics. Data was captured at two moments during the interaction: "pre-demanding" includes actions before the SP demanded the assigned medicine, and "post-demanding" includes all actions by the completion of the visit. We analyze N=200 pre-demanding and N=200 post-demanding observations for the childhood diarrhea case scenario.

- Table 2, last 4 rows: it appears to me that the order of magnitude for these is off (unless perhaps these numbers are per patient?).

Thank you for raising this. We examined this more carefully and have updated the table with data presented from 2019 when this SP study was conducted in parallel with a clinic survey at the same clinic sample. (The previous table 3 contained data from the AHME program evaluation from 2014.)

- Table 3, notes: sentence "Single observations..." is unclear.

Thank you. We agree that this sentence is unclear. We have removed this sentence.

The purpose of the deleted sentence was to describe the merge of the SP data to the provider data – in that if different SP observations were seen by the same provider, the provider would show up in the same frequency. However, since there is one observation per clinic, it made sense to us to remove this unclear sentence.

- Missing observations: please add a note on the missing observations, e.g. for "any antibiotic" and "any anti parasitic" (Tables 3, B1, B2).

We have examined comments from the supervisors and SPs for the visits that correspond to these missing observations. All SPs who conducted these visits had conversations with providers at the clinic but did not receive any medicines. The observations have been recoded to "0" referring to not having received each medicine.

- Typo on p.12, l.15: our study is about the overprescription and overuse of antimalarials in Mali.

Thank you for pointing this out, and we apologize for misspeaking. We have updated this to say that the findings reported by Lopez et al. (2020) “assessed whether patients’ demands influence overprescription and overuse of antimalarials in Mali” (p. 11, lines 32-33).

- Appendix Figure A.1 has a couple of typos in the first box.

Thank you for catching these. We reviewed the content in all of the boxes and also ran the text through spellcheck to correct typos.

References:

Thank you for sharing the references below! They were either already included or now have been included in the current submission.

Navneet Aujla, Yen-Fu Chen, Yasara Samarakoon, Anna Wilson, Natalia Grolmusová, Abimbola Ayorinde, Timothy P Hofer, Frances Griffiths, Celia Brown, Paramjit Gill, Christian Mallen, Jo Sartori, Richard J Lilford, Comparing the use of direct observation, standardized patients and exit interviews in low- and middle-income countries: a systematic review of methods of assessing quality of primary care, *Health Policy and Planning*, Volume 36, Issue 3, April 2021, Pages 341–356, [https://urldefense.com/v3/__https://doi.org/10.1093/heapol/czaa152__;!!LQC6Cpwp!8KW-vtmUXAr7mJwyLI7PfjSRit6QM6-PxpSPneVVLo-MCsew8Os-BJv0kHx3gc_C\\$](https://urldefense.com/v3/__https://doi.org/10.1093/heapol/czaa152__;!!LQC6Cpwp!8KW-vtmUXAr7mJwyLI7PfjSRit6QM6-PxpSPneVVLo-MCsew8Os-BJv0kHx3gc_C$) .

Janet Currie, Wanchuan Lin, Juanjuan Meng, Addressing antibiotic abuse in China: An experimental audit study, *Journal of Development Economics*, Volume 110, 2014, Pages 39-51, [https://urldefense.com/v3/__https://doi.org/10.1016/j.jdeveco.2014.05.006__;!!LQC6Cpwp!8KW-vtmUXAr7mJwyLI7PfjSRit6QM6-PxpSPneVVLo-MCsew8Os-BJv0kGqyiB66\\$](https://urldefense.com/v3/__https://doi.org/10.1016/j.jdeveco.2014.05.006__;!!LQC6Cpwp!8KW-vtmUXAr7mJwyLI7PfjSRit6QM6-PxpSPneVVLo-MCsew8Os-BJv0kGqyiB66$) .

Janet Currie, Wanchuan Lin, Wei Zhang, Patient knowledge and antibiotic abuse: Evidence from an audit study in China, *Journal of Health Economics*, Volume 30, Issue 5, 2011, Pages 933-949, [https://urldefense.com/v3/__https://doi.org/10.1016/j.jhealeco.2011.05.009__;!!LQC6Cpwp!8KW-vtmUXAr7mJwyLI7PfjSRit6QM6-PxpSPneVVLo-MCsew8Os-BJv0kFzPzHYS\\$](https://urldefense.com/v3/__https://doi.org/10.1016/j.jhealeco.2011.05.009__;!!LQC6Cpwp!8KW-vtmUXAr7mJwyLI7PfjSRit6QM6-PxpSPneVVLo-MCsew8Os-BJv0kFzPzHYS$) .

Global Antibiotic Resistance Partnership—Kenya Working Group. 2011. Situation Analysis and Recommendations: Antibiotic Use and Resistance in Kenya. Washington, DC and New Delhi: Center for Disease Dynamics, Economics & Policy
[https://urldefense.com/v3/__https://cddep.org/wp-content/uploads/2017/08/garp-kenya_sa.pdf__;!!LQC6Cpwp!8KW-vtmUXAr7mJwyLI7PfjSRit6QM6-PxpSPneVVLo-MCsew8Os-BJv0kNqzresT\\$](https://urldefense.com/v3/__https://cddep.org/wp-content/uploads/2017/08/garp-kenya_sa.pdf__;!!LQC6Cpwp!8KW-vtmUXAr7mJwyLI7PfjSRit6QM6-PxpSPneVVLo-MCsew8Os-BJv0kNqzresT$)

Goldberg, Pinelopi Koujianou, 1996. “Dealer Price Discrimination in New Car Purchases: Evidence from the Consumer Expenditure Survey” *Journal of Political Economy*, Volume 104, Number 3, [https://urldefense.com/v3/__https://doi.org/10.1086/262035__;!!LQC6Cpwp!8KW-vtmUXAr7mJwyLI7PfjSRit6QM6-PxpSPneVVLo-MCsew8Os-BJv0kAiQ4Xng\\$](https://urldefense.com/v3/__https://doi.org/10.1086/262035__;!!LQC6Cpwp!8KW-vtmUXAr7mJwyLI7PfjSRit6QM6-PxpSPneVVLo-MCsew8Os-BJv0kAiQ4Xng$)

National Policy for the Prevention and Containment of Antimicrobial Resistance, Nairobi, Kenya: Government of Kenya, April 2017. [https://urldefense.com/v3/__https://www.health.go.ke/wp-content/uploads/2017/04/Kenya-AMR-Containment-Policy-_Final_April.pdf__;!!LQC6Cpwp!8KW-vtmUXAr7mJwyLI7PfjSRit6QM6-PxpSPneVVLo-MCsew8Os-BJv0kJNymhA7\\$](https://urldefense.com/v3/__https://www.health.go.ke/wp-content/uploads/2017/04/Kenya-AMR-Containment-Policy-_Final_April.pdf__;!!LQC6Cpwp!8KW-vtmUXAr7mJwyLI7PfjSRit6QM6-PxpSPneVVLo-MCsew8Os-BJv0kJNymhA7$)

Heckman, James J. 1998. "Detecting Discrimination." *Journal of Economic Perspectives*, 12 (2): 101-

116.
DOI: 10.1257/jep.12.2.101

Reviewer 2 comments and author responses:

Dr. Xiaohui Wang, Lanzhou University

Comments to the Author:

This study examined the role of patient demand for inappropriate care using the methods of USP and vignette data. Inappropriate care refers to the behavior of the providers. That is to say, whether the provider prescribes or dispenses the antibiotic amoxicillin or the deworming drug albendazole. This paper makes a valuable contribution to the study of quality of care using the SP method.

Here are some Minor comments:

1. The format: please standardize the reference format in the text. (P10, Line 5 reference 23,23,34 vs. Line9 reference 21,23

Thank you for bringing this to our attention. We have redone the references and the reference formatting is now standardized throughout the text.

2. Inconsistent page numbers throughout the text

We have double checked the page numbers and have also changed the page number formatting of the supplemental file to S1-S22 (corresponding to pages 1 through 22) so that they are not confused with the main text pagination. We have maintained our line numbers associated with our file so that we can refer to where we made corrections more easily.

3. Please follow the CONSORT reporting guideline to organize the manuscript.

We have worked to incorporate several aspects that were missing based on the CONSORT reporting guidelines to organize the manuscript, including Figure 1. We have also added sample size calculations to Appendix A (pp. S11-12) and Appendix Table A2. We note that our study is a randomized experiment and not a randomized trial so no follow-up period exists. We have also uploaded the CONSORT reporting guideline checklist.

We have updated the Figure 1 based on the Reviewer's suggestion to follow the CONSORT flow diagram as closely as possible. The updated figure is now:

4. Some comment regarding the main text are as follows:

P2 Abstract

Methods

L19. According to the manuscript, I assume you used unannounced SP visits instead of SP visits? Would you please clarify?

Yes, we apologize for the lack of clarity. We did utilize unannounced SP visits. We have clarified this in the title, abstract, and throughout the text.

Results

Line 28 of this study aims to examine the effect of quality of care when patients demand different types of inappropriate medicines. But according to the result, the outcome is dispensing rate of the two types of medicine. Is there any possibility to measure the quality of care more specifically?

Thank you for this comment. We have reviewed this and updated the abstract and discussion text to more clearly reflect that the objective of our study was to examine the role of patient demand for inappropriate care on prescribing and dispensing practices for childhood diarrhea in Kenya.

The abstract's results reads (p. 1):

Neither significantly changed any correct management outcomes, such as treatment or referral elsewhere.

We have also added the underlined text to the outcomes description in the methods section (p. 7, lines 18-19):

Using SP and vignette data, we constructed binary measures for our main outcomes of interest: correct case management and whether any unnecessary medicines were prescribed or dispensed, since one aspect of quality of care is not only dispensing correct medicines but also not dispensing inappropriate medicines.

Is there any underlining incentive-induced difference if the patient requires the medicine at a different time point?

Thank you for this question. For this study, we assume the different times are balanced across each demanding arm. We have chosen to include the following text on p. S11 (3rd full paragraph) of the Appendix:

It is quite possible that demanding a medicine when the provider is writing a prescription or about to dispense drugs could have an underlining incentive-induced difference. In this study, we assume that the different time points for demanding are balanced across each demanding arm.

Because of the reviewer's comment, we added "only" to clarify the following sentence in Data Sources in the Methods section (p. 6, line 21). Unfortunately, we are not able to ascertain from the data how frequently this occurred.

SP requests were done at the end of the visit or earlier only if it was necessary to avoid an unusual interaction.

P6 Line 35. is the number (36) typo? Or does this refer to reference 36?

We thank the reviewer for bringing this to our attention. "(36)" refers to reference 36. Also, we have redone the references and the reference formatting is now standardized throughout the text.

Methods
Data Sources

P37 Line 22. The SPs were locally recruited and trained in January 2019. The last day of the training was 01-Feb-19, while the visits were conducted from March 8 to May 28. One concern is that after two months, the SPs might forget some of the scripts? Any quality control before the fieldwork?

Refresher training, ongoing piloting (check)

Quality control is an important concern for any fieldwork including the SP method, so we appreciate the reviewer for raising this. Between the last day of training and the first day in the data, the team conducted a 2-week pilot, followed by 1-week of classroom training on the final cases. We actually began fieldwork on February 28, but the team halted SP fieldwork because of an unrelated issue. It then re-commenced in March 8.

We recognize the concern raised by the reviewer and thus have decided to expand the description of the pilot and motivation for experiments in Appendix A3.3 on p. S9. The text now reads:

Between February 5-15, 2019, the SPs piloted in Nairobi, and some teams also traveled out to three different areas in Kenya to ensure that we understood whether the experiments for the case needed to be adapted for different regions (since the clinic sample was spread across the country).

Given the experience during the pilot, we designed experiments for demanding unnecessary medicines. Figure A6 shows the case scenario narratives with scripts for the two experiments: demanding amoxicillin and demanding albendazole.

After the pilot and between fieldwork, the supervisors conducted refresher trainings in the classroom on the cases and did quality checks on the programmed SP exit questionnaire. Throughout fieldwork, the supervisors also conducted sessions where the case and experiments were reviewed again as a team to ensure there was no evolution of presentation in any given SP.

P37 Line 24. This study described three time points when the SP was assigned to the “demanding dawa ya minyao” group. Will the first one affect the providers' behavior? Say, change the prescribe itself?

P37 Line 39 The same concern.

Thank you for this question. It is quite possible that demanding a medicine when the provider is writing a prescription or about to dispense drugs could have an underlining incentive-induced difference. Examining this, though very interesting, is outside the scope of this particular study. We assume that the different time points for demanding are balanced across each demanding arm. (Unfortunately, we are not able to ascertain from the data how frequently each occurred.) Another study could very interestingly examine these incentives more scientifically.

Because of the reviewer's comment, we added "only" to clarify the following sentence in Data Sources in the Methods section (p. 6, line 20).

SP requests were done at the end of the visit or earlier only if it was necessary to avoid an unusual interaction.

Also, we have chosen to include the following text on p. S10 (2nd full paragraph) of the Appendix:

When we began piloting the demanding experiment before fieldwork, we did not have the first two time points ((i) when the provider writes a prescription or is about to dispense drugs, (ii) when the provider asks what the patient wants). We only had the third (at the end of the interaction). However, the pilot anecdotally demonstrated to us that some providers did (i) and (ii) in the same moment, and for the SPs, it was unusual and out of their character to not respond if they came in "wanting the medicine they demanded".

P8 Line 8. It is a good way to conduct the provider survey among those who saw the SPs to explore the know-do gap.

Thank you.

Outcomes

P8 Line 14. What is the difference between prescribe and dispense?

We use 'prescribe' to capture any situation where the provider may have written a prescription, including when the SP may not have actually received the medicine (e.g., if there was a stockout). Similarly, we use 'dispense' to capture any situation where the provider may have given the medicine, including when the provider may not have written a prescription. Together, 'dispense/prescribe' allows us to capture the intent of the provider on giving the medicine to the patient, regardless of whether the SP walked away with it.

Because of the reviewer's question, we have chosen to include the following text on p. 7 (lines 27-32) in the outcomes subsection of methods for clarity:

We define 'prescribe/dispense' as a term to capture the intent of the provider on giving the medicine to the patient, regardless of whether the SP walked away with it: 'prescribe' can capture a situation where the provider may have written a prescription, including when the SP may not have actually received the medicine (e.g., a stockout) and 'dispense' captures a situation where the provider may have given the medicine, including when the provider may not have written a prescription.

Results

P9 Line 27. why measure the person waiting in the waiting room when the SP arrived?

We measure the number of people waiting when the SP arrives to capture how busy the clinic was. Because we do not utilize administrative data, we do not have medical records (which may not exist in this setting and if they did exist, they may not be of the same quality) to summarize utilization numbers. By measuring the patients in the waiting room, we have a proxy for utilization.

Because of the reviewer's questions, we have chosen to include red text below on p. 8 (lines 38-39) for clarity:

On average, there were approximately 1.55 (95% CI: 1.16-1.94) individuals waiting in the waiting room when the SP arrived (to capture how busy the clinic was in lieu of utilization data)...

Discussion

P10 Line 16-30. the authors described the result of anecdotal narratives during debriefs with the supervisors of the SP fieldwork. It might be easier to understand the method used in this study. If I were the author, I would like reorganize this section. To be specify, move these to "results" as part of the qualitative research method.

We thank the reviewer for this comment. After some deliberation, we have decided to remove the anecdotal narratives from this paper, since this study did not have a qualitative component with methods rigorous enough to present. We keep in the discussion that providers may be trading off clinical benefits and risks with profits, but instead of supporting it with the narratives, we suggest that a future study could examine this in more depth. We felt that adding it to the results as part of the qualitative research method was outside the scope of the original research question for this manuscript.

P19. I failed to find Figure 1. Clinic sample and SP randomized study design.

We include the original Figure 1 from our submission for the Reviewer:

We note for the reviewer that we have updated the Figure 1 based on the Reviewer's suggestion to follow CONSORT reporting guidelines. The updated figure is now:

P22. In Table 3. Summary statistics of SP visits, under “Visit characteristics”, it seems that there are some indicators with missing data. e.g. “Minutes spent with provider”, just wondering, is this due to SP failed to collect this information? Or other reason?

Thank you for raising this data issue. From your comment, we examined the 11 observations which were consistently missing data (because of a skip pattern in the questionnaire). We examined the comments entered by the supervisors and SPs for the visits that correspond to the 11 missing observations (which we will note is a small percentage of the total sample; 5.5% of total observations). All the missing observations had conversations with staff or providers at the clinic but did not receive any medicines. Minutes spent with the provider and other variables have been recoded (values highlighted) to match the comments.

This has now been corrected in Table 3 for the majority of the variables on pp. 22-23. For the other variables:

- Provider is female: we were not able to recover this variable for 4 observations
- Provider age group: the data was not entered for the 11 visits below
- Provider qualification: the data was not entered for the 11 visits below
- Provider knowledge of diarrhea correct management: of the 200 visits, we were only able to return and find providers corresponding to 140 of the visits.
- Provider did a good job explaining: the data was not entered for the 11 visits below

Clinic Index	Form	Demanding experiment	Comments	Provider is female	Time with provider (min)	Asked to return	Referred elsewhere
1	8557	Demanding amoxicillin	This clinic that i visited was closed. The provider usually treats people from her home and she is very cautious about who she treats. So when Itried to inquire at her home I was told that she had travelled to western kenya and was not available at that time and that is the same thing she (the provider) had told my colleague the previous day. She looked very suspicious	1	0	0	0
2	5348	Demanding albendazole	None		0	0	0
3	4582	Demanding albendazole	Once I arrived at the facility, i registered my daughter's details. The nurses that were to conduct the triage however insisted that without my daughter the doctor would not see me. So they said I should go get her bring her back.		0	1	0
4	5279	Demanding amoxicillin	The provider politely turned me down and told me to go and bring the child for treatment.		1	1	0
5	5414	Demanding albendazole	The provider was very concerned about my case but he politely asked me to go and bring the child for him to be able to treat. He said it is very risky to treat a child who is absent.	0	1	1	0
6	5403	Demanding albendazole	The doctor was really concerned but remained firm that I should go and bring the child for treatment.		1	0	1
7	5274	Demanding amoxicillin	only (i.e. while the patient is absent), he can only prescribe for oral medicine and may be the child might be requiring IV fluids. He said diarrhoea is one of leading cause of death for children and I should rush home and take the baby to a nearby hospital.	0	1	0	1
8	5284	Demanding amoxicillin	The provider was concerned but politely told me that he could not treat a child in absences. He said that if it were a grown-up he would have considered but for a child it is tricky.	0	1	0	1
9	4537	Demanding albendazole	When I told the provider that I didn't have the child. He asked where the child was and told me he could not treat a child who is not there since there are tests that need to be done to a diarrhoea child e.g. see the level of dehydration and also run some lab tests (stool tests) to see the source of the problem. He added that it was illegal to treat inabsencia patient and advised me to take the child to a nearby clinic.	0	1	0	1
10	5368	Demanding albendazole	The provider insisted that I take my child to a nearby clinic mostly a public facility. She politely declined to treat a child who was absent.	1	1	1	1
11	5335	Demanding albendazole	The provider was very firm in his decision not to treat in absencia child and told me to go and take the child to a hospital. He did not want to hear my pleading.	0	1	1	1

Reviewer: 1

Competing interests of Reviewer: None.

Reviewer: 2

Competing interests of Reviewer: There are no competing interests.

Editor(s)' Comments to Author (if any):

VERSION 2 – REVIEW

REVIEWER	Sautmann, Anja World Bank Group, Development Economics Research Group
REVIEW RETURNED	25-Feb-2022

GENERAL COMMENTS	I would like to thank the authors for addressing all my requests carefully and comprehensively. I enjoyed reading this new version of the manuscripts and have no further comments.
---

REVIEWER	Wang, Xiaohui Lanzhou University, School of Public Health
REVIEW RETURNED	23-Feb-2022

GENERAL COMMENTS	Thank you for your very professional work, I enjoyed reading the revised manuscript. The majority of the comments and concerns from my side were well addressed. here are two further minor suggestions  1. Figure 1 could be further improved. How about combining the content of excluded reasons with the successful visits to make it more clear? 2. I assume an arrow pointing from the data collection frame to analysis was missing. This paper undoubtedly adds new knowledge of health services in the real healthcare setting in Kenya. Thanks to the author and her team for sharing their experience and founding using USP method to access the health workers' behavior.
--